# Generalized Category Discovery Utilizing Reciprocal Learning and Class-wise Distribution Regularization

## Abstract

Generalized Category Discovery (GCD) aims to identify unlabeled samples by leveraging the base knowledge from labeled ones, where the unlabeled set consists of both base and novel classes. Since clustering methods are time-consuming at inference, parametric-based approaches have become more popular. However, recent parametric-based methods suffer inferior base discrimination due to the unreliable self-supervision. To address this issue, we propose a Reciprocal Learning Framework (RLF) that introduces an auxiliary branch devoted to base classification. During training, the main branch filters the pseudo-base samples to the auxiliary branch. In response, the auxiliary branch provides more reliable soft labels for the main branch, leading to a virtuous cycle. Furthermore, we introduce Class-wise Distribution Regularization (CDR) to mitigate the leaning bias towards base classes. CDR essentially increases the prediction confidence of the unlabeled data and boosts the novel class performance. Combined with both components, our method achieves superior performance in all classes with negligible extra computation. Extensive experiments on seven GCD datasets validate the effectiveness of our method, *e.g.* delivering a notable 2.1% improvement on the Stanford Cars dataset. Our codes will be available upon acceptance.

## 1 Introduction

With the development of deep learning in recent years, models can perform well in traditional tasks such as image recognition He et al. (2016; 2017); Vaswani (2017); Dosovitskiy (2020). Generally, the models rely on abundant annotated data in a closed scenario where the unlabeled data share the same classes with the labeled training data. However, these models have limitations in the real-world scenario where unlabeled data comes from unknown classes. In this way, Category Discovery (CD) has garnered attention in the machine learning community. Initially, Han et al. (2019) proposes Novel Class Discovery (NCD) which is designed to cluster novel class data with the assistance of labeled data exclusively. However, NCD assumes the unlabeled data all belong to novel classes, which is unrealistic in practical scenarios. Recently, Generalized Category Discovery (GCD) Vaze et al. (2022) has emerged and it allows the unlabeled data spanning both base and novel categories. Compared to the NCD task, GCD is more practical and challenging in real-world scenarios.

Vaze et al. (2022) first defined the GCD problem and tackled it using contrastive learning along with the semi-supervised $k$-means clustering method. Wen et al. (2023)further proposes an effective parametric framework SimGCD which outperforms the clustering methods with reduced inference time. Due to its effectiveness, the parametric framework has become popular in GCD research. Wang et al. (2024) designs a two-stage framework on the pre-trained SimGCD model that introduces both global and spatial prompts to fine-tune the model. Cao et al. (2024) observes that SimGCD suffers catastrophic forgetting of base classes and they propose a novel regularization to address it. Despite the significant advancements in parametric methods, experimental results show that these methods often suffer inferior base discrimination. To quantitatively reveal the main limitation in existing works, we define the *oracle base accuracy* for evaluating base discrimination, which solely considers base-class prediction, and calculates the accuracy of unlabeled base data. Fig. 1 validates that SimGCD and LegoGCD lag behind the supervised-only reference (SupRef) which exclusively

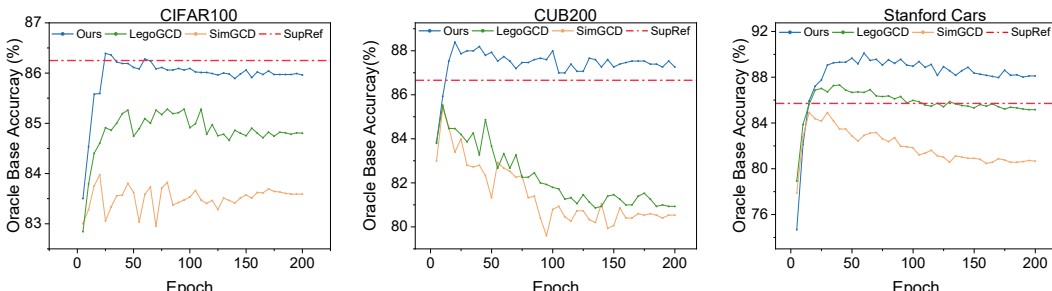

Figure 1: Comparison of the oracle base class accuracy between SimGCD, LegoGCD, and our method. SimGCD and LegoGCD exhibit poor performance, falling behind the supervised reference (SupRef). Contrarily, our method exhibits enhanced discrimination, even surpassing SupRef.

utilizes labeled data for training. This disparity primarily arises from the unreliable soft labels in self-supervised learning.

To promote base discrimination, we design a reciprocal learning framework (RLF). In particular, we insert an auxiliary token named AUX in the model architecture. This AUX is concatenated with the CLS token and image feature tokens to form the input of the final block. Subsequently, the corresponding AUX output is dedicated to a base-only classifier while the CLS output is designated for the all-class classifier. During training, the main branch filters pseudo-base samples, which are predicted to the base classes, and directs them to the auxiliary branch. In feedback, the auxiliary branch provides reliable base class distribution to the main branch. This collaboration between the two branches contributes to more robust base predictions, improving base-class discrimination and overall accuracy.

However, the reciprocal framework may incur learning bias toward base classes that more novel samples are misclassified into the base classes. To alleviate the above bias, we propose a Class-wise Distribution Regularization (CDR) loss. Specifically, CDR involves calculating the expected distribution for each category based on mini-batch predictions. Then, CDR loss promotes expectation consistency between two views of mini-batch and boosts prediction confidence. Since each class can be treated equally, CDR effectively mitigates the bias and boosts novel class performance. By integrating CDR into the RLF, our method further obtains improved performance.

Our key contributions can be summarized as follows: (1) We define the oracle base class accuracy to evaluate the base discrimination of GCD models and unravel the inferior discrimination in parametric methods. (2) We design a novel reciprocal framework to promote base class discrimination and a class-wise distribution regularization loss to improve novel class performance. (3) We conduct comprehensive evaluations of our method on seven GCD datasets, where it significantly outperforms state-of-the-art approaches in most cases.

## 2 RELATED WORKS

**Semi-Supervised Learning (SSL)** is a prominent area in machine learning that addresses the challenge of training models with limited labeled data. Pseudo Label Lee et al. (2013) iteratively assigns pseudo labels for unlabeled data, which join the labeled set for further training. Mean-teacher Tarvainen & Valpola (2017), UDA Xie et al. (2020), Fixmatch Sohn et al. (2020) adopt confidence threshold to generate pseudo labels on weak augmented samples and utilize it to supervise strongly augmented samples, and DST Chen et al. (2022) proposes an adversary framework to refine pseudo labels. Consistency-based methods introduce extra regularization on unlabeled data. ConMatch Kim et al. (2022) adds self-supervised features regularization while SimMatch extends consistency to the semantic and instance levels. PAWS Assran et al. (2021) incorporates self-supervised clustering principles into SSL, learning better representation. Several works explore more realistic scenarios including open-word semi-supervised learning Cao et al. (2022), and long-tailed semi-supervised learning Wei & Gan (2023). Prevailing semi-supervised methods widely adopt threshold-based pseudo-label learning during training. However, this mechanism faces significant limitations when unlabeled data include samples from unknown classes.

**Novel Class Discovery (NCD)** aims to recognize novel classes in unlabeled data, by exploiting knowledge from known classes. Han et al. (2019) first proposes the NCD problem and addresses it utilizing a two-stage training strategy. Han et al. (2020) employ rank statistics to find positive data pairs and pull them closer. OpenMix Zhu et al. (2023) generates virtual samples by MixUp between labeled and unlabeled data, guiding the model to resist nosily labeled data. Zhong et al. (2021) proposes neighborhood contrastive leaning to aggregate pseudo-positive pairs. Fini et al. (2021) introduces a unified objective framework with the Sinkhorn-Knopp algorithm, allowing cross-entropy to operate on both labeled and unlabeled sets. Current NCD methods often utilize the Optimal-Transport (OT) algorithm to dynamically cluster unlabeled data. However, OT achieves unsatisfactory novel class performance when unlabeled data involves known class samples. Therefore, current NCD methods are unsuitable for the GCD task. Moreover, CRNCD Gu et al. (2023) introduces a two-stage class-relationship distillation approach to improve novel-class performance. However, we observe that this distillation shows inferior performance on GCD. Unlike CRNCD, we propose a novel one-stage distillation method tailored for GCD.

**Generalized Category Discovery (GCD)** is to cluster unlabeled images by leveraging the base knowledge from labeled images, where the unlabeled set compromises both base and novel classes. Vaze et al. (2022) formulates the GCD problem and conducts contrastive training on a pre-trained ViT model Dosovitskiy (2020) with DINO Caron et al. (2021), clustering the data using semi-supervised $k$-means. CiPR Hao et al. (2024) designs a novel contrastive learning method by exploiting cross-instance positive relations in labeled data and introducing a hierarchical clustering algorithm. PromptCAL Zhang et al. (2023) designs a two-stage framework that exploits affinity graphs to enhance semantic discrimination. GPC Zhao et al. (2023) applies Gaussian mixture models that learn robust representation and estimate the novel class number. InfoSieve Rastegar et al. (2024) and CMS Choi et al. (2024) achieve great clustering results utilizing specialized contrastive learning. Wen et al. (2023) proposes a parametric framework that trains a prototype classifier to fit all categories. SimGCD utilizes mean-entropy regularization to automatically find novel classes. As SimGCD boots GCD performance with lower inference latency, the parametric framework becomes popular. SPTNet Wang et al. (2024) introduces a two-stage strategy that combines the global and spatial prompts to further finetune the SimGCD model. LegoGCD Cao et al. (2024) finds SimGCD suffers catastrophic forgetting in training and solves it by adding regularization to potential known class samples. While parametric-based methods achieve great GCD performance, they often suffer degraded base discrimination. To address this issue, we propose a Reciprocal Learning Framework (RLF) that provides more reliable base pseudo-labels and effectively strengthens base performance with negligible extra computation cost. Combined with class-wise distribution regularization, our method achieves superior performance.

## 3 METHOD

### 3.1 PRELIMINARIES

**Problem Formulation.** Generalized Category Discovery (GCD) aims to adaptively cluster unlabeled data utilizing the knowledge from labeled data. GCD is built upon the open-world dataset, which compromises two subsets: labeled dataset $\mathcal{D}^l = \{(\boldsymbol{x}_i, y_i)\} \in \mathcal{X} \times \mathcal{Y}^l$ and unlabeled dataset $\mathcal{D}^u = \{(\boldsymbol{x}_i, y_i)\} \in \mathcal{X} \times \mathcal{Y}^u$. Formally, $\mathcal{Y}^l$ is a subset of $\mathcal{Y}^u$, and $\mathcal{Y}^u$ spans all categories. Following previous research, the number of $|\mathcal{Y}^u|$ is assumed as the prior. GCD adopts a transductive training strategy in which all the samples are involved in the training process.

**Parametric Clustering.** Wen et al. (2023) proposes an efficient parametric framework that builds a prototype classifier for clustering. Specifically, the classifier weight is the set of prototypes $\mathcal{C} = \{\boldsymbol{c}_1, \ldots, \boldsymbol{c}_K\}$, where $K$ is the total number of prototypes. Given an image $\boldsymbol{x}_i$, the model correspondingly output feature $f(\boldsymbol{x}_i)$, and the probability of category $k$ is denoted as:

$$\boldsymbol{p}_i^{(k)} = \frac{\exp\left(\cos\left(f\left(\boldsymbol{x}_i\right), \boldsymbol{c}_k\right)/\tau_s\right)}{\sum_{k'} \exp\left(\cos\left(f\left(\boldsymbol{x}_i\right), \boldsymbol{c}_{k'}\right)/\tau_s\right)}, \tag{1}$$

where $\cos$ denotes the cosine similarity between two vectors and $\tau_s$ is the temperature scalar. Similarly, the shrink probability $\boldsymbol{q}_i$ can be derived by substituting $\tau_s$ with a smaller $\tau_t$. Subsequently, SimGCD adopts the cross entropy loss $\mathcal{L}_{\text{ce}}(\boldsymbol{q}, \boldsymbol{p}) = -\sum_k \boldsymbol{q}^{(k)} \log \boldsymbol{p}^{(k)}$ to regularize the probability

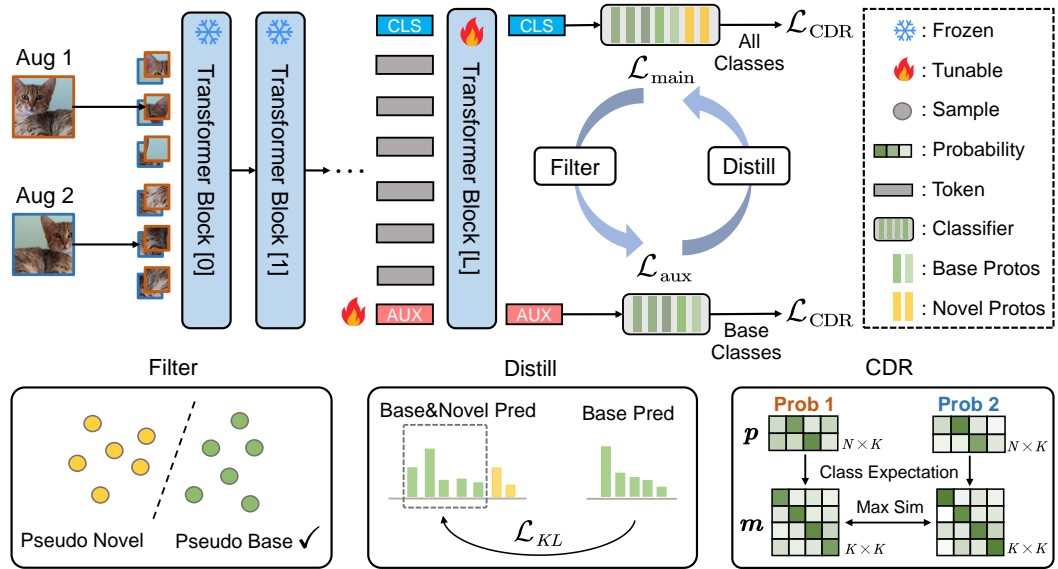

Figure 2: Overview of our method. We insert an auxiliary token AUX before the last block of the ViT backbone. The final AUX feature is utilized for the base-only classifier while the CLS feature is assigned to the all-class classifier. The main branch filters the pseudo-base samples to the aux branch for better base class learning. In response, the auxiliary branch provides the main branch with refined base class distribution. Class-wise Distribution Regularization (CDR) boosts novel performance by maximizing the similarity between class-wise probability matrices $\boldsymbol{m}$ from two views.

self-consistency between two views of an image. For $i$-th image, the loss is formulated as:

$$\mathcal{L}_{\text{self}}^{(i)} = \frac{1}{2}\mathcal{L}_{\text{ce}}\left(\boldsymbol{q}_i', \boldsymbol{p}_i\right) + \frac{1}{2}\mathcal{L}_{\text{ce}}\left(\boldsymbol{q}_i, \boldsymbol{p}_i'\right), \tag{2}$$

where $\boldsymbol{p}_i'$ and $\boldsymbol{q}_i'$ are the prototype probabilities of another view. Additionally, SimGCD employs a mean-entropy maximization regulariser for clustering: $H(\overline{\boldsymbol{p}}) = -\sum_k \overline{\boldsymbol{p}}^{(k)}\log\overline{\boldsymbol{p}}^{(k)}$ where $\overline{\boldsymbol{p}}$ is the mean predicted probability of all the samples. The supervised loss for the labeled data is the sum of cross-entropy and supervised contrastive learning losses Khosla et al. (2020):

$$\mathcal{L}_{\text{sup}}^{(i)} = \mathcal{L}_{\text{ce}}\left(\boldsymbol{y}_i, \boldsymbol{p}_i\right) + \frac{1}{|\mathcal{P}_i|}\sum_{q\in\mathcal{P}_i} -\log\frac{\exp\left(\cos(f\left(\boldsymbol{x}_i\right), f(\boldsymbol{x}_q'))\,/\tau_c\right)}{\sum_{n\neq i}\exp\left((f\left(\boldsymbol{x}_i\right), f\left(\boldsymbol{x}_n'\right))\,/\tau_c\right)}, \tag{3}$$

where $\boldsymbol{y}_i$ is the one-hot distribution associated with $y_i$, $\tau_c$ denotes the temperature scalar for supervised contrastive learning and the $\mathcal{P}_i$ is the positive index set sharing the same label as $\boldsymbol{x}_i$. While SimGCD applies InfoNCE Oord et al. (2018) loss in the training, we found the loss tends to push apart same-class features, which conflicts with the SupCon loss and impairs feature discrimination. Consequently, we chose to remove InfoNCE in our approach to maintain better class discrimination. Overall, parametric clustering loss $\mathcal{L}_{\text{cls}}$ is the average per-sample combination of supervised loss, self-consistency loss, and entropy regularization loss:

$$\mathcal{L}_{\text{cls}} = \lambda\mathcal{L}_{\text{sup}} + (1-\lambda)(\mathcal{L}_{\text{self}} - \epsilon H(\overline{\boldsymbol{p}})), \tag{4}$$

where $\lambda$ is the balance weight belonging to [0,1] and $\epsilon$ is the scalar to control entropy regularization.

## 3.2 Reciprocal Learning Framework

**Motivation.** While SimGCD demonstrates greater effectiveness than clustering methods, it falls short in base class discrimination. Specifically, when focusing on base classification, the unlabeled base data is defined as $\mathcal{D}_{\text{base}}^u = \{(\boldsymbol{x}_i, y_i) | (\boldsymbol{x}_i, y_i) \in \mathcal{D}^u, y_i \in \mathcal{Y}^l\}$. The oracle base accuracy is defined as $ACC_{\text{OB}} = \frac{1}{|\mathcal{D}_{\text{base}}^u|}\sum_{\boldsymbol{x}_i, y_i \in \mathcal{D}_{\text{base}}^u} \mathbb{1}\left(\widetilde{y}_i = y_i\right)$, where $\widetilde{y}_i$ is the predicted base class result. As depicted in Fig. 1, prevailing parametric methods exhibit unsatisfactory oracle base accuracy, falling

behind the supervised-only reference. We conduct a detailed loss analysis of the parametric baseline. Specifically, N2N represents the percentage of novel class samples classified as novel, while B2B indicates the percentage of base class samples classified as base.

As shown in Table R1, applying $\mathcal{L}_{\text{sup}}$ exclusively builds a strong discriminative model as the training data is correctly labeled. Since the novel class prototype is randomly initialized, all unlabeled data would be classified as the base. Combining $\mathcal{L}_{\text{sup}}$ and $\mathcal{L}_{\text{self}}$, novel data are still misclassified due to the absence of annotations. Additionally, $\mathcal{L}_{\text{self}}$ adopts a shrink-soft probability distillation mechanism that increases the incorrect confidence of novel data, which

Table R1: Effect of loss configuration.

|  | OB(%) | N2N(%) | B2B(%) |
|---|---|---|---|
| $\mathcal{L}_{\text{sup}}$ | 86.7 | 0 | 100 |
| $\mathcal{L}_{\text{sup}} + \mathcal{L}_{\text{self}}$ | 85.6 | 0 | 100 |
| $\mathcal{L}_{\text{sup}} + H$ | 84.4 | 18.8 | 98.1 |
| $\mathcal{L}_{\text{sup}} + \mathcal{L}_{\text{self}} + H$ | 83.9 | 93.1 | 69.8 |

harms model representation and degrades OB. The combination of $\mathcal{L}_{\text{sup}}$ and $H$ increases novel class prediction, slightly improving N2N but causing a decline in OB. This decline occurs because $H$ encourages a balanced average prediction. When all three losses are applied, the model predicts more novel class samples, leading to 93.1% N2N. This is because $\mathcal{L}_{\text{self}}$ increases the novel class confidence, facilitating novel class prototype learning. However, $H$ simultaneously drives the misclassification of some base samples, while $\mathcal{L}_{\text{self}}$ amplifies incorrect novel confidence, further deteriorating the pseudo-label quality. These findings suggest that both $\mathcal{L}_{\text{self}}$ and $H$ contribute to unreliable pseudo-labels. Besides, $\mathcal{L}_{\text{self}}$ is particularly susceptive to $H$ and exacerbates noisy learning. This motivates us to introduce a more robust base-class expert to assist $\mathcal{L}_{\text{self}}$. By providing more reliable base labels, this intervention aims to enhance discrimination and mitigate the side effects of $H$.

To this end, we propose a one-stage reciprocal learning framework. As shown in Fig. 2, we insert the auxiliary token AUX before the last block, concatenating it with CLS and feature tokens to form the input. The ultimate AUX feature is utilized for the base-only classification while CLS feature is assigned to the all-class classifier. Different from the CLS feature, which is unique to each image, the AUX token is a trainable parameter shared across all training samples.

During the training procedure, the main branch is akin to generic parametric clustering. Besides, the main branch filters the pseudo-base class samples to the auxiliary branch according to the prediction result, *i.e.*, if a sample is predicted to belong to the base classes, it will also be involved in the auxiliary branch. In response, the auxiliary branch distills the base class prediction of pseudo-base samples to the main branch. The collaboration between the two branches effectively enhances base discrimination, mitigates the influence of noise labels, and facilitates the model in acquiring improved representations.

Note that most of the training samples will be predicted as the base classes in the initial, the auxiliary branch also incorporates novel samples. To this end, the auxiliary branch adopts self-supervised learning and supervised learning, rather than threshold-based semi-supervised methods. Furthermore, we utilize the maximum probability as the uncertainty weight for each pseudo-base sample in the cross-branch distillation. The distillation loss for a pseudo-base sample $i$ is denoted as:

$$\mathcal{L}_{\text{dis}}^{(i)} = \max(\boldsymbol{p}_{b,i}^{\text{aux}}) \cdot \mathcal{L}_{\text{KL}}(\boldsymbol{p}_{b,i}^{\text{aux}}, \boldsymbol{p}_{b,i}), \tag{5}$$

where $\boldsymbol{p}_b^{\text{aux}}$, $\boldsymbol{p}_b$ is the base class distribution from the auxiliary and main branch, $\mathcal{L}_{\text{KL}}$ is the standard KL-divergence loss, and the auxiliary probability is detached in the distillation. Consequently, the loss functions of the two branches can be presented as:

$$\mathcal{L}_{\text{main}} = \mathcal{L}_{\text{cls}} + \alpha \mathcal{L}_{\text{dis}}, \ \mathcal{L}_{\text{aux}} = \mathcal{L}_{\text{sup}} + \mathcal{L}_{\text{self}}. \tag{6}$$

where $\alpha$ is the scalar weight to control the distillation strength.

### 3.3 CLASS-WISE DISTRIBUTION REGULARIZATION.

While the proposed reciprocal framework can effectively improve base class discrimination, it still shows inferior performance in the novel classes. This is primarily due to the cross-branch distillation being confined to base class distributions, resulting in a learning bias where training samples are more likely to be recognized as base classes. Fig. 3(a) shows that the predicted novel samples lag behind the ground truth number. To mitigate the learning bias, we propose a novel class-wise distribution regularization (CDR) shown in the bottom right of Fig. 2.

Adopted by Zhang et al. (2024), the class-wise expected distribution $\boldsymbol{m}$ for category $k$ is:

$$\boldsymbol{m}_k = \frac{1}{\sum_{i=1}^{N} \boldsymbol{p}_i^{(k)}} \left( \sum_{i=1}^{N} \boldsymbol{p}_i^{(k)} \boldsymbol{p}_i \right), \tag{7}$$

where $N$ is the batch size, and $\boldsymbol{m}$ has a dimension of $K \times K$. For the main branch, $K^{\text{main}}$ is the number of all classes, while $K^{\text{aux}}$ denotes the number of base classes in the auxiliary branch. Intuitively, the $k$-th probability of $m_k$, denoted as $m(k, k)$, reflects the confidence that "the mini-batch contains at least one sample belonging to category $k$."

**Theorem 1.** *The sum of all elements in $\boldsymbol{m}_k$ equals 1, i.e., $\mathbf{1}^T \boldsymbol{m}_k = 1$ Zhang et al. (2024).*

*Proof.* Please refer to the Appendix A.1.

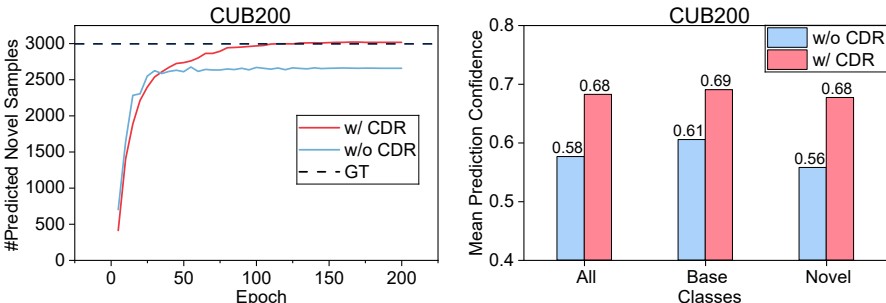

Figure 3: The efficacy of CDR is evident in two aspects. Left: CDR induces more predicted novel class samples. Right: CDR contributes to higher prediction confidence.

Theorem 1 shows that $\boldsymbol{m}_k$ conforms to the standard probability distribution. Intuitively, the class-wise prediction should be consistent between the two views of the images and close to the one-hot distribution. To this end, for class $k$, the CDR loss is formulated as

$$\mathcal{L}_{\text{CDR}}^{(k)} = 1 - \langle \boldsymbol{m}_k, \boldsymbol{m}_k' \rangle, \tag{8}$$

where $\langle \cdot, \cdot \rangle$ denotes the inner product calculation, representing the similarity between two distributions, and $\boldsymbol{m}_k'$ is the expectation from another view. Since each class is treated equally, CDR effectively alleviates the bias towards the base classes in the main branch.

**Theorem 2.** $\mathcal{L}_{\text{CDR}}^{(k)}$ *equals to zero $\iff$ $\boldsymbol{m}_k$ equals $\boldsymbol{m}_k'$ and is a one-hot distribution.*

*Proof.* Please refer to the Appendix A.1.

Theorem 2 indicates CDR essentially increases the prediction confidence, approaching the one-hot distribution. The effectiveness of CDR is evidenced in Fig. 3, as it leads to a higher number of predicted novel class samples, reducing learning bias and boosting prediction confidence. Furthermore, the CDR loss operates independently of ground-truth labels and is compatible with both branches. When applied to the auxiliary branch, CDR also benefits base class learning with minimal impact on the novel class performance of the main branch. By integrating the CDR loss into the reciprocal framework, the overall loss is summarized as:

$$\mathcal{L} = \mathcal{L}_{\text{main}} + \mathcal{L}_{\text{aux}} + \beta \mathcal{L}_{\text{CDR}}, \tag{9}$$

where $\beta$ is the control factor to assign the regularization weight. After the training procedure, we abandon the auxiliary classifier and only keep the main branch for evaluation. As a result, the inference latency difference from SimGCD is negligible.

## 4 EXPERIMENTS

### 4.1 EXPERIMENTAL SETUP

**Datasets.** Following previous works, we evaluate our method on seven different GCD datasets. Those consist of generic image recognition datasets CIFAR10/100 Krizhevsky et al. (2009) and

Table 1: Comparative results on the Semantic Shift Benchmark and Herbarium-19.

| Methods | CUB200 | | | Stanford Cars | | | FGVC-Aircraft | | | Herbarium-19 | | |
|---|---|---|---|---|---|---|---|---|---|---|---|---|
| | All | Base | Novel | All | Base | Novel | All | Base | Novel | All | Base | Novel |
| $k$-means Macqueen (1967) | 34.3 | 38.9 | 32.1 | 12.8 | 10.6 | 13.8 | 16.0 | 14.4 | 16.8 | 13.0 | 12.2 | 13.4 |
| RS+ Han et al. (2021) | 33.3 | 51.6 | 24.2 | 28.3 | 61.8 | 12.1 | 26.9 | 36.4 | 22.2 | 27.9 | 55.8 | 12.8 |
| UNO+ Fini et al. (2021) | 35.1 | 49.0 | 28.1 | 35.5 | 70.5 | 18.6 | 40.3 | 56.4 | 32.2 | 28.3 | 53.7 | 14.7 |
| ORCA Cao et al. (2022) | 35.3 | 45.6 | 30.2 | 23.5 | 50.1 | 10.7 | 22.0 | 31.8 | 17.1 | 20.9 | 30.9 | 15.5 |
| *CRNCD Gu et al. (2023) | 62.7 | 71.6 | 58.2 | 54.1 | 75.7 | 43.7 | 54.4 | 59.5 | 51.8 | 41.3 | 60.7 | 30.9 |
| GCD Vaze et al. (2022) | 51.3 | 56.6 | 48.7 | 39.0 | 57.6 | 29.9 | 45.0 | 41.1 | 46.9 | 35.4 | 51.0 | 27.0 |
| DCCL Pu et al. (2023) | 63.5 | 60.8 | 64.9 | 43.1 | 55.7 | 36.2 | - | - | - | - | - | - |
| GPC Zhao et al. (2023) | 55.4 | 58.2 | 53.1 | 42.8 | 59.2 | 32.8 | 46.3 | 42.5 | 47.9 | 36.5 | 51.7 | 27.9 |
| PromptCAL[1] Zhang et al. (2023) | 51.1 | 55.4 | 48.9 | 42.6 | 62.8 | 32.9 | 44.5 | 44.6 | 44.5 | 37.0 | 52.0 | 28.9 |
| SimGCD Wen et al. (2023) | 60.3 | 65.6 | 57.7 | 53.8 | 71.9 | 45.0 | 54.2 | 59.1 | 51.8 | 44.0 | 58.0 | 36.4 |
| CMS Choi et al. (2024) | 68.2 | 76.5 | 64.0 | 56.9 | 76.1 | 47.6 | 56.0 | **63.4** | 52.3 | 36.4 | 54.9 | 26.4 |
| InfoSeive Rastegar et al. (2024) | 69.4 | **77.9** | 65.2 | 55.7 | 74.8 | 46.4 | 56.3 | 63.7 | 52.5 | 41.0 | 55.4 | 33.2 |
| SPTNet Wang et al. (2024) | 65.8 | 68.8 | 65.1 | 59.0 | **79.2** | 49.3 | 59.3 | 61.8 | 58.1 | 43.4 | 58.7 | 35.2 |
| LegoGCD Cao et al. (2024) | 63.8 | 71.9 | 59.8 | 57.3 | 75.7 | 48.4 | 55.0 | 61.5 | 51.7 | 45.1 | 57.4 | 38.4 |
| **Ours** | **69.5** | 76.4 | **65.9** | **61.1** | 76.6 | **53.6** | **60.6** | 62.2 | **59.8** | **46.4** | 61.2 | **38.4** |

ImageNet-100 Tian et al. (2020); Semantic Shit Benchmark (SSB) Vaze et al. (2021) datasets: CUB200 Wah et al. (2011), Stanford Cars Krause et al. (2013), and FGVC-Aircraft Maji et al. (2013); large-scale fine-grained dataset: Herbarium-19 Tan et al. (2019). Formally, each dataset is partitioned into base and novel subsets. The novel subset data is entirely unlabeled, while half of the base data is labeled during training, with the remaining half left unlabeled. For a fair comparison, we adopt the same random seed in the data split with Vaze et al. (2022).

**Evaluation Metric.** We adopt cluster accuracy ($ACC$) to evaluate the performance of our method. More specifically, given the samples' prediction $\hat{y}$ and ground-truth labels $y$, the Hungarian optimal assignment algorithm Kuhn (1955) allocates the clustering result and calculates the accuracy. $ACC = \frac{1}{|\mathcal{D}^u|} \sum_{i=1}^{|\mathcal{D}^u|} \mathbb{1}(y_i = \mathcal{G}(\hat{y}_i))$, where $\mathcal{G}$ denotes the optimal permutation function.

**Implementation details.** In alignment with other methods, we conduct our experiments using the pre-trained DINO ViT-B/16 backbone Caron et al. (2021), fine-tuning only the last block and the auxiliary token across all datasets. The final output retains the features from the CLS and AUX tokens for classification. The default learning rate is set to 0.1, following a cosine annealing decay schedule. The model is trained for 200 epochs with a batch size of 128. Following Wen et al. (2023), the temperature scalars are $\tau_c = 0.1, \tau_s = 0.07$, while $\tau_t$ scaling from 0.07 to 0.04 within 30 epochs, and the balance weight $\lambda = 0.35$. The hyper-parameters in our method are specified as $\alpha = 0.5, \beta = 0.5$. The default augmentation includes Resize, RandomCrop, Random Horizontal Flip, Color Jittering, and Image Normalization. All experiments are conducted on a single NVIDIA GeForce 3090 GPU based on PyTorch.

## 4.2 MAIN RESULTS

We compare our approach with SOTA methods including clustering-based methods: $k$-means Macqueen (1967), GCD Vaze et al. (2022), GPC Zhao et al. (2023), PromptCAL Zhang et al. (2023), DCCL Pu et al. (2023), InfoSieve Rastegar et al. (2024), CMS Choi et al. (2024); parametric-based methods: SimGCD Wen et al. (2023), SPTNet Wang et al. (2024), LegoGCD Cao et al. (2024) and strong baseline derived from NCD: RS+ Han et al. (2021), UNO+ Fini et al. (2021), ORCA Cao et al. (2022), CRNCD Gu et al. (2023). The best results are highlighted in bold and * denotes reproduced results.

**Evaluation on fine-grained datasets.** Table 1 shows the comparative results on four fine-grained datasets which are more challenging than the generic. Clustering methods demonstrate inadequate performance, falling far behind the parametric methods. Our method consistently outperforms the others, achieving 2.1% on Stanford Cars, 1.3% on Aircraft, and 1.3% on Herbarium-19. Specifically, we obtain the highest novel class accuracy in all four datasets as well as leading in base accuracy than parametric based methods in most cases. The great performance in both base and novel classes aligns with our expectations.

Table 2: Comparative results on generic image recognition datasets.

| Methods | CIFAR10 | | | CIFAR100 | | | ImageNet-100 | | |
|---|---|---|---|---|---|---|---|---|---|
| | All | Base | Novel | All | Base | Novel | All | Base | Novel |
| $k$-means Macqueen (1967) | 83.6 | 85.7 | 82.5 | 52.0 | 52.2 | 50.8 | 72.7 | 75.5 | 71.3 |
| RS+ Han et al. (2021) | 46.8 | 19.2 | 60.5 | 58.2 | 77.6 | 19.3 | 37.1 | 61.6 | 24.8 |
| UNO+ Fini et al. (2021) | 68.6 | **98.3** | 53.8 | 69.5 | 80.6 | 47.2 | 70.3 | 95.0 | 57.9 |
| ORCA Cao et al. (2022) | 81.8 | 86.2 | 79.6 | 69.0 | 77.4 | 52.0 | 73.5 | 92.6 | 63.9 |
| *CRNCD Gu et al. (2023) | 96.9 | 97.5 | 96.6 | 80.3 | 84.7 | 71.5 | 81.4 | 94.4 | 74.8 |
| GCD Vaze et al. (2022) | 91.5 | 97.9 | 88.2 | 73.0 | 76.2 | 66.5 | 74.1 | 89.8 | 66.3 |
| DCCL Pu et al. (2023) | 96.3 | 96.5 | 96.9 | 75.3 | 76.8 | 70.2 | 80.5 | 90.5 | 76.2 |
| GPC Zhao et al. (2023) | 92.2 | 98.2 | 89.1 | 77.9 | 85.0 | 63.0 | 76.9 | 94.3 | 71.0 |
| PromptCAL[1] Zhang et al. (2023) | 97.1 | 97.7 | 96.7 | 76.0 | 80.8 | 66.6 | 75.4 | 94.2 | 66.0 |
| SimGCD Wen et al. (2023) | 97.1 | 95.1 | 98.1 | 80.1 | 81.2 | 77.8 | 83.0 | 93.1 | 77.9 |
| InfoSieve Rastegar et al. (2024) | 94.8 | 97.7 | 93.4 | 78.3 | 82.2 | 70.5 | 80.5 | 93.8 | 73.8 |
| CMS Choi et al. (2024) | - | - | - | 82.3 | **85.7** | 75.5 | 84.7 | **95.6** | 79.2 |
| SPTNet Wang et al. (2024) | 97.3 | 95.0 | **98.6** | 81.3 | 84.3 | 75.6 | 85.4 | 93.2 | 81.4 |
| LegoGCD Cao et al. (2024) | 97.1 | 94.3 | 98.5 | 81.8 | 81.4 | **82.5** | 86.3 | 94.5 | 82.1 |
| **Ours** | **97.4** | 96.4 | 97.9 | **82.8** | 84.1 | 80.1 | **86.5** | 93.9 | **82.8** |

**Evaluation on generic datasets.** As shown in Table 2, we present the comparison on generic datasets including CIFAR10/100 and ImageNet-100. Our method consistently achieves the best performance in all classes, with a 0.5% improvement on CIFAR100 to CMS. Our method still surpasses the two-stage method SPTNet and with even fewer parameters in the training. Note that, DINO is pre-trained on the extensive generic dataset ImageNet-1000 Deng et al. (2009). Leveraging the robust feature representation of DINO, existing methods exhibit comparable performance on datasets like CFAR10 and ImageNet-100. Furthermore, the three datasets are rich in labeled data for training, resulting in minimal degradation in discrimination for parametric methods. While our method may not show substantial enhancements, it consistently delivers the best results.

## 4.3 VISUALIZATION

As shown in Fig. 4, we utilize t-SNE Van der Maaten & Hinton (2008) to visualize the feature distribution between DINO, SimGCD, LegoGCD, and our model on the FGVC-Aircraft dataset. The t-SNE results for DINO demonstrate poor clustering performance, primarily due to the significant domain gap between the ImageNet and Aircraft datasets, which hinders effective feature learning. Meanwhile, SimGCD and LegoGCD achieve unsatisfactory feature clustering. It is observed that "Class 3" features of SimGCD spread out in the feature space, while LegoGCD forms two clusters of "Class 3". In contrast, our model reveals distinct clusters corresponding to different categories. The visualization comparison validates the superior feature representation of our model.

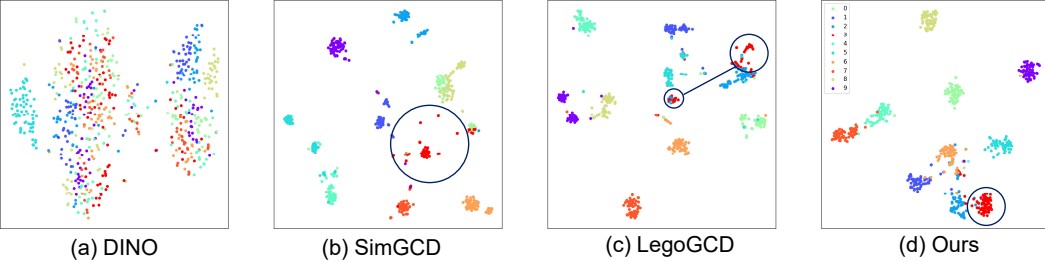

|     (a) DINO     |     (b) SimGCD     |     (c) LegoGCD     |     (d) Ours     |

Figure 4: T-SNE visualization comparing DINO, SimGCD, LegoGCD, and our method on the FGVC-Aircraft dataset, with samples randomly selected from 10 classes.

## 4.4 ABLATION STUDY

**Effect of different loss components.** As previously outlined, our approach mainly has four loss components: main branch loss (Main), auxiliary branch loss (AUX), cross-branch distillation (Dis-

till), and class-wise distribution regularization (CDR). Here we demonstrate their effectiveness on CIFAR100 and CUB200 datasets. Additionally, we introduce the oracle base class accuracy (OB) as a reference. Table 3 shows the GCD performance with different loss configurations. Specifically, using only the Main loss in (1) serves as the baseline, while (5) represents our complete method. The comparison of (1) with (2) indicates that CDR enhances overall performance with minimal influence on OB, contributing substantial improvements in novel classes—3.3% on CIFAR100 and 5.3% on CUB200. The inclusion of AUX in training (3) leads to comprehensive improvement and better discrimination. Since the last block is shared between the two branches, the model learns more robust parameters, resulting in enhanced feature representation. Comparing (3) with (4), we find that distillation further strengthens performance, improving base accuracy by 0.5% on CIFAR100 and 2.2% on CUB200. Finally, combining all components in (5) results in further improvement in novel class performance while preserving base class accuracy, ultimately yielding the best overall results. When either the Distill (6) or the AUX (7) is exclusively removed, our model suffers a significant drop in base performance. This outcome highlights the necessity of both components in our approach. Comparing (5) and (8) shows that CDR in the both branches helps improve base class performance without exacerbating the base class learning bias.

Table 3: Ablation experiments on different configurations of loss components: Main, AUX, Distill, and CDR. OB denotes the Oracle base class accuracy and $M$ represents only on main-branch.

| | Main | AUX | Distill | CDR | CIFAR100 | | | | CUB200 | | | |
|---|---|---|---|---|---|---|---|---|---|---|---|---|
| | | | | | All | Base | Novel | OB | All | Base | Novel | OB |
| (1) | ✓ | | | | 79.6 | 82.6 | 73.6 | 84.4 | 62.1 | 70.8 | 57.7 | 83.9 |
| (2) | ✓ | | | ✓ | 80.4 | 82.2 | 76.9 | 84.8 | 65.7 | 71.1 | 63.0 | 83.7 |
| (3) | ✓ | ✓ | | | 80.3 | 82.9 | 75.2 | 85.3 | 64.3 | 73.7 | 59.7 | 85.9 |
| (4) | ✓ | ✓ | ✓ | | 80.8 | 83.8 | 74.7 | **86.2** | 66.5 | 75.9 | 61.8 | 87.2 |
| (5) | ✓ | ✓ | ✓ | ✓ | **82.8** | **84.1** | **80.1** | 86.0 | **69.5** | **76.4** | **65.9** | **87.3** |
| (6) | ✓ | ✓ | | ✓ | 81.6 | 82.7 | 79.2 | 85.5 | 67.6 | 71.7 | 65.6 | 85.8 |
| (7) | ✓ | | ✓ | ✓ | 81.0 | 81.9 | 79.2 | 84.9 | 65.9 | 73.8 | 61.9 | 84.6 |
| (8) | ✓ | ✓ | ✓ | $M$ | 82.3 | 83.5 | 80.0 | 85.6 | 68.7 | 75.3 | 65.4 | 86.7 |

**Effect of different $\alpha$ and $\beta$.** As indicated in Equations (6) and (9), we utilize $\alpha$ and $\beta$ to control the distillation and regularization strength. Fig. 5 (a) illustrates the GCD performance curves with varying values of $\alpha$. As $\alpha$ increases, we observe a significant improvement in base class accuracy, aligning with the intuition that stronger distillation enhances base class dissemination. However, when $\alpha$ becomes excessively large, it leads to degradation in novel class performance, ultimately harming overall accuracy. As shown in Fig. 5 (b), we see that increasing the weight of $\beta$ notably impacts novel class performance. However, overly large $\beta$ shows a negative effect on base class accuracy. Our analysis indicates that the optimal values for $\alpha$ and $\beta$ are approximately 0.5, which yields the best overall performance.

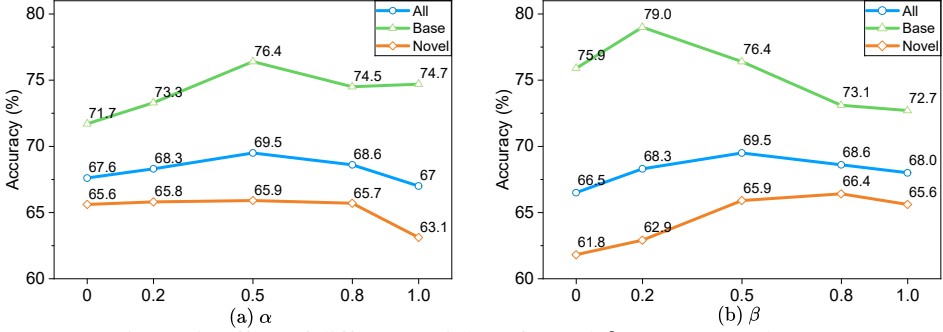

Figure 5: Effect of different weights of $\alpha$ and $\beta$ on CUB200 dataset.

**Effect of different regularization.** We here present the GCD performance across different probability regularized methods. The baseline is our reciprocal learning framework and the comparative methods include entropy minimization(ENT) Grandvalet & Bengio (2004), minimum class confusion (MCC) Jin et al. (2020), maximizing $F$-norm of probability matrix (BNM) Cui et al. (2020),

and label-encoding risk minimization (LERM) Zhang et al. (2024). Besides, we modify the CDR loss into pair-wise distribution regularization (PDR) that directly maximizes the probability similarity between two views of a sample.

Table 4 indicates that prevailing regularization methods are not competitive in the GCD task. ENT shows a serious negative effect, leading to considerable degradation in both base and novel classes. MCC effectively improves the base performance yet harms the novel performance in CIFAR100 and FGVC-Aircraft. BNM and LERM show marginal effects as their performance is close to baseline. While PDR benefits all three datasets, the improvement is slight and widens the performance difference between the base and novel classes in FGVC-Aircraft. In contrast, our proposed CDR effectively boosts novel class accuracy while maintaining base performance. Overall, the comparison demonstrates our proposed CDR is more appropriate for GCD tasks.

Table 4: Comparison of different regularization on CIFAR100, CUB200, and FGVC-Aircraft.

| Methods | CIFAR100 | | | CUB200 | | | FGVC-Aircraft | | |
|---|---|---|---|---|---|---|---|---|---|
| | All | Base | Novel | All | Base | Novel | All | Base | Novel |
| RLF | 80.8 | 83.8 | 74.7 | 66.5 | 75.9 | 61.8 | 56.6 | 61.8 | 54.0 |
| + ENT Grandvalet & Bengio (2004) | 72.3 | 80.5 | 56.0 | 62.2 | 66.0 | 60.3 | 51.7 | 52.2 | 51.5 |
| + MCC Jin et al. (2020) | 79.4 | **84.6** | 68.9 | 67.2 | **77.6** | 62.0 | 56.3 | 63.4 | 52.7 |
| + BNM Cui et al. (2020) | 80.2 | 83.9 | 72.7 | 65.9 | 76.1 | 60.8 | 55.8 | 60.4 | 53.6 |
| + LERM Zhang et al. (2024) | 80.5 | 83.2 | 75.2 | 65.6 | 76.4 | 60.2 | 56.2 | 64.2 | 52.2 |
| + PDR | 81.0 | 83.7 | 75.5 | 68.2 | 76.5 | 64.0 | 57.2 | **64.5** | 53.5 |
| + CDR | **82.8** | 84.1 | **80.1** | **69.5** | 76.4 | **65.9** | **60.6** | 62.2 | **59.8** |

**Effect of estimated category number.** As the previous evaluation is built on the known category numbers $K$, we here report the results with estimated categories borrowed from off-the-shelf methods GCD Vaze et al. (2022) and GPC Zhao et al. (2023). As shown in Table 5, our method exhibits reduced performance on ImageNet-100 and Stanford Cars under GCD estimation, yet still surpasses other methods. With the increased number of prototypes, the mean-entropy regularization renders unlabeled samples clustered into more clusters. Consequently, all three datasets maintain the base class performance yet suffer novel class performance degradation. Particularly, The impact on CUB200 is minimal, with a 1.0% degradation. By leveraging the advanced estimated algorithm within GPC, the performance gap is narrowed across both datasets, with differences of merely 0.2%, 0.1%, and 0.4% among the three datasets. The result indicates that our approach is not reliant on exact category numbers.

Table 5: Comparison of estimated category numbers on ImageNet-100, CUB200, and Stanford Cars.

| Methods | $K$ | ImageNet-100 | | | CUB200 | | | Stanford Cars | | |
|---|---|---|---|---|---|---|---|---|---|---|
| | | All | Base | Novel | All | Base | Novel | All | Base | Novel |
| Ours | 100 /200/196 | 86.5 | 93.9 | 82.8 | 69.5 | 76.4 | 65.9 | 61.1 | 76.6 | 53.6 |
| GCD Vaze et al. (2022) | 109 / 231 /230 | 73.8 | 92.1 | 64.6 | 49.2 | 56.2 | 46.3 | 36.3 | 56.6 | 25.9 |
| SimGCD Wen et al. (2023) | 109 / 231/230 | 81.1 | 90.9 | 76.1 | 61.0 | 66.0 | 58.6 | 49.1 | 65.1 | 41.3 |
| SPTNet Wang et al. (2024) | 109 / 231/230 | 83.4 | 91.8 | 74.6 | 65.2 | 71.0 | 62.3 | - | - | - |
| **Ours** | 109 / 231/230 | **84.4** | **93.2** | **80.0** | **68.4** | **77.1** | **64.1** | **58.6** | **76.4** | **50.8** |
| GPC Zhao et al. (2023) | 103 / 212/201 | 75.3 | 93.4 | 66.7 | 52.0 | 55.5 | 47.5 | 38.2 | 58.9 | 27.4 |
| **Ours** | 103 / 212/201 | **86.3** | **94.1** | **82.4** | **69.4** | **78.5** | **64.8** | **60.7** | **77.7** | **52.5** |

## 5 CONCLUSION

In this paper, we propose a novel approach for enhancing generalized category discovery performance. To promote base class discrimination, we design a Reciprocal Learning Framework (RLF) that incorporates an auxiliary branch, which generates reliable soft labels for the main branch, while the main branch effectively filters pseudo-base samples based on an all-class classifier prediction. Additionally, to mitigate the learning bias towards base classes, we introduce Class-wise Distribution Regularization (CDR), which significantly boosts the prediction confidence of unlabeled data and improves novel class performance. These two components are complementary, and their integration leads to great performance of both base and novel classes. Extensive experiments demonstrate the superiority of our method, achieving state-of-the-art results across seven GCD datasets.

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

## A  APPENDIX

### A.1  THEORETICAL SUPPORT.

**Proof of Theorem 1**

*Proof.* $\mathbf{1}^T \boldsymbol{m}_k = \frac{\mathbf{1}^T}{\sum_{i=1}^N \boldsymbol{p}_i^{(k)}} \left( \sum_{i=1}^N \boldsymbol{p}_i^{(k)} \boldsymbol{p}_i \right) = \frac{\left( \sum_{i=1}^N \boldsymbol{p}_i^{(k)} (\mathbf{1}^T \boldsymbol{p}_i) \right)}{\sum_{i=1}^N \boldsymbol{p}_i^{(k)}} = \frac{\sum_{i=1}^N \boldsymbol{p}_i^{(k)}}{\sum_{i=1}^N \boldsymbol{p}_i^{(k)}} = 1.$

**Proof of Theorem 2**

*Proof.* Let $\mathbf{a}$ and $\mathbf{b}$ be two probability distribution vectors in $\mathbb{R}^n$ to present $\boldsymbol{m}_k$ and $\boldsymbol{m}_k'$ :

$$\mathbf{a} = [a_1, a_2, \ldots, a_n], \quad \mathbf{b} = [b_1, b_2, \ldots, b_n]$$

subject to the constraints:

$$\sum_{i=1}^n a_i = 1, \quad \sum_{i=1}^n b_i = 1, \quad a_i \geq 0, \quad b_i \geq 0 \text{ for } i = 1, 2, \ldots, n.$$

The inner product is given by:

$$\mathbf{a} \cdot \mathbf{b} = \sum_{i=1}^n a_i b_i.$$

By the Cauchy-Schwarz inequality, we have:

$$\left( \sum_{i=1}^n a_i b_i \right)^2 \leq \left( \sum_{i=1}^n a_i^2 \right) \left( \sum_{i=1}^n b_i^2 \right).$$

Since $\sum_{i=1}^n a_i = 1$ and $\sum_{i=1}^n b_i = 1$, we can observe:

$$\sum_{i=1}^n a_i^2 \leq \sum_{i=1}^n a_i = 1, \sum_{i=1}^n b_i^2 \leq \sum_{i=1}^n b_i = 1.$$

Thus, we have:

$$\left( \sum_{i=1}^n a_i b_i \right)^2 \leq 1 \cdot 1 = 1 \implies \sum_{i=1}^n a_i b_i \leq 1.$$

For equality $\sum_{i=1}^n a_i b_i = 1$ to hold, the Cauchy-Schwarz inequality must achieve equality, which occurs if and only if $a_i$ and $b_i$ are linearly dependent:

$$a_i = c b_i \text{ for some constant } c \text{ for all } i.$$

Given the constraints $\sum_{i=1}^n a_i = 1$ and $\sum_{i=1}^n b_i = 1$, it follows that:

$$1 = c \sum_{i=1}^n b_i = c \cdot 1 \implies c = 1.$$

Therefore, we have:

$$a_i = b_i \text{ for all } i.$$

Besides,

$$\sum_{i=1}^n a_i^2 \leq \sum_{i=1}^n a_i = 1 \implies a_i \in \{0, 1\}$$

which means:

$$a_j = 1 \text{ for some } j \text{ and } a_i = 0 \text{ for } i \neq j.$$

Thus, we conclude that:

$$\mathbf{a} = \mathbf{b} \text{ and both are one-hot distributions.}$$

## A.2 DATASET SPLIT

As shown in Fig. 6, we illustrate the dataset split of generalized category discovery(GCD) and compare it with semi-supervised learning (SSL) and novel class discovery (NCD). SSL assumes the labeled and unlabeled data share the same classes, NCD suggests unlabeled data all form the novel classes, while GCD allows unlabeled data belonging to all classes. The comparison indicates GCD task is more challenging and practical in real-world scenarios.

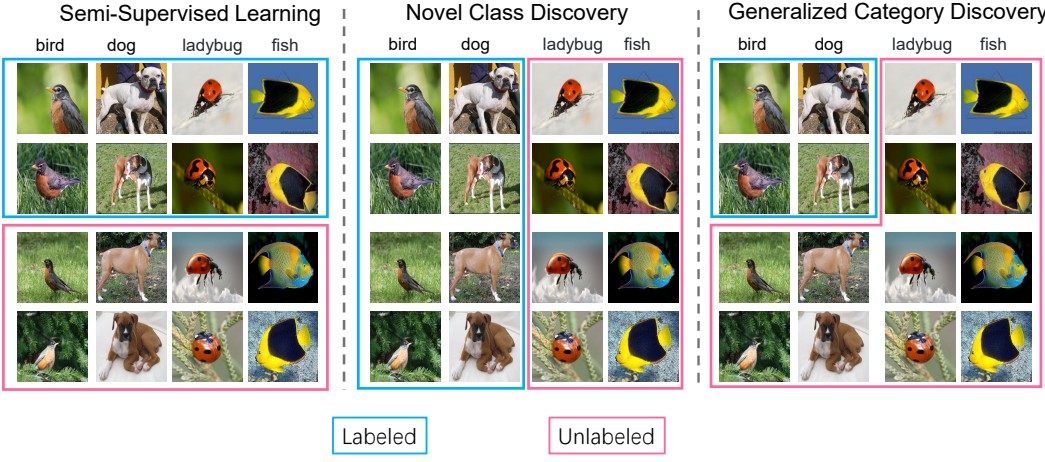

Figure 6: Difference of dataset split among SSL, NCD, and GCD.

## A.3 LOSS ANALYSIS

Fig. 7 shows SimGCD retains a high supervised cross-entropy (SupCE) loss during training, which indicates the noise labels in SimGCD. In contrast, our model achieves a near-zero SupCE loss. Since we introduce an auxiliary branch, it can provide more reliable soft labels to the main branch. This effectively eliminates noisy information and enhances discrimination capabilities.

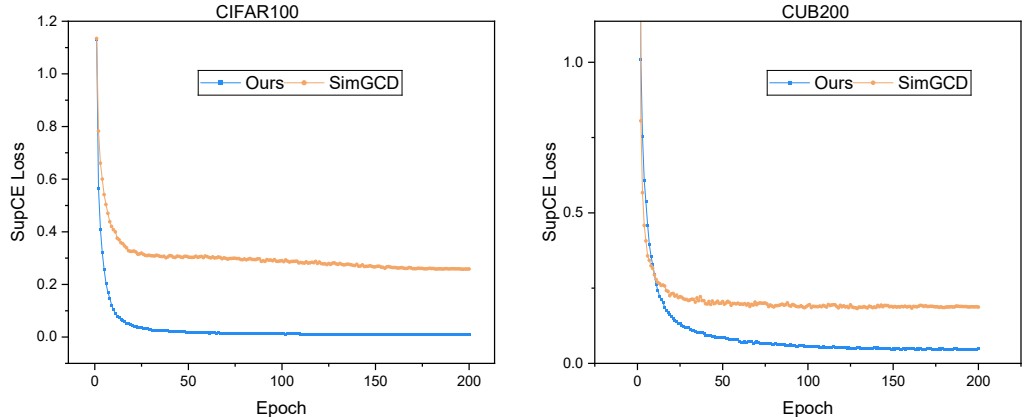

Figure 7: Supevised cross-entropy descent loss curves on CIFAR100 and CUB200.

### A.4 PARAMETER ANALYSIS

We provide an overview of the parameter quantities in parametric models in Table 6. Despite incorporating an additional base classifier in the auxiliary branch, our method excludes the projector, resulting in significant parameter savings. The token's contribution to the overall model size is minimal, enabling us to utilize the fewest parameters during training. During the evaluation, we abandon the base classifier and retain the extra token with the backbone, which has a negligible parameter overhead.

Table 6: Parameter quantity statistics among parametric models.

| Methods | Backbone | | Classifier | | Projector | | Extra | | Total |
|---|---|---|---|---|---|---|---|---|---|
| | Name | #Param. | Name | #Param. | Name | #Param. | Name | #Param. | |
| SimGCD | ViT-B/16 | 85,798,656 | All | 153,600 | MLP | 6,295,808 | None | 0 | 92,248,064 |
| LegoGCD | ViT-B/16 | 85,798,656 | All | 153,600 | MLP | 6,295,808 | None | 0 | 92,248,064 |
| SPTNet | ViT-B/16 | 85,798,656 | All | 153,600 | MLP | 6,295,808 | Prompts | 105,120 | 92,353,184 |
| Ours | ViT-B/16 | 85,798,656 | All+Base | 230,400 | None | 0 | Token | 768 | 86,029,824 |

### A.5 COMPARISON OF ORACLE BASE ACCURACY

As shown in Table 7, the auxiliary branch AUX in our model achieves the best oracle base accuracy in most cases, indicating superior base class discrimination. As training advances, AUX effectively guides the main branch CLS to improve discrimination. Consequently, CLS outperforms SimGCD and LegoGCD in most cases.

Table 7: Oracle Base accuracy comparison on seven GCD datasets. CLS and AUX denote the main and auxiliary branches.

| | CIFAR10 | CIFAR100 | ImageNet-100 | CUB200 | Stanford Cars | FGVC-Aircraft | Herbarium-19 |
|---|---|---|---|---|---|---|---|
| SupRef | **98.7** | 86.3 | 95.4 | 86.7 | 85.7 | 72.5 | 65.4 |
| SimGCD | 98.4 | 83.6 | 95.4 | 80.5 | 80.7 | 72.8 | 68.6 |
| LegoGCD | 98.4 | 84.7 | 95.5 | 80.9 | 85.5 | 70.4 | 70.2 |
| Ours-CLS | 98.4 | 86.0 | 95.5 | 87.3 | 88.1 | 75.5 | 76.3 |
| Ours-AUX | 98.6 | **86.4** | **96.0** | **88.1** | **89.9** | **75.9** | **76.4** |

### A.6 COMPARISON WITH DIFFERENT UNCERTAINTY WEIGHTS.

As depicted in Equation (5), we adopt the maximum probability in the auxiliary branch $\max\left(\boldsymbol{p}_b^{\mathrm{aux}}\right)$to denote the uncertainty weight of the pseudo-base samples. Here we make a comparison with different uncertainty weights. $\boldsymbol{c}_{\max}$ is the prototype associated with the maximum probability. When the uncertainty weight is set to 0, distillation is excluded, resulting in reduced base accuracy. Conversely, the weight of 1 biases the model towards base classes, impairing novel class performance. Using the maximum cosine similarity for uncertainty yields similar results to using the maximum probability. Additionally, the uncertainty weight in the auxiliary branch obtains better performance, suggesting its greater reliability compared to the main branch.

Table 8: Comparison of different uncertainty weights.

| Uncertainty weight | CIFAR100 | | | CUB200 | | |
|---|---|---|---|---|---|---|
| | All | Base | Novel | All | Base | Novel |
| 0 | 81.4 | 83.2 | 77.9 | 68.4 | 74.3 | 65.4 |
| 1 | 81.6 | **85.0** | 75.0 | 67.8 | 76.7 | 63.4 |
| $\cos\left(f\left(\boldsymbol{x}_i\right),\boldsymbol{c}_{\max}\right)$ | 82.0 | 83.4 | 79.0 | 68.6 | **76.9** | 64.5 |
| $\cos\left(f^{\mathrm{aux}}\left(\boldsymbol{x}_i\right),\boldsymbol{c}_{\max}^{\mathrm{aux}}\right)$ | 82.6 | 84.2 | 79.2 | 69.2 | 76.8 | 65.3 |
| $\max\left(\boldsymbol{p}_b\right)$ | 82.1 | 83.4 | 79.3 | 69.1 | 76.5 | 65.4 |
| $\max\left(\boldsymbol{p}_b^{\mathrm{aux}}\right)$ | **82.8** | 84.1 | **80.1** | **69.5** | 76.4 | **65.9** |

## A.7 Discussion with CRNCD

As CRNCD Gu et al. (2023) and our approach both involve distillation, there are several key differences, listed below.

- **Different tasks.** CRNCD aims to deal with novel class discovery (NCD), where all unlabeled data belong to novel classes. In contrast, our focus is on GCD, where unlabeled data comprises both novel and base classes. Due to the intrinsic difference between these two tasks, CRNCD demonstrates unsatisfactory performance in GCD.
- **Different motivations for using distillation.** The distillation in CRNCD aims to improve novel class performance, whereas our distillation is intended to promote base class discrimination.
- **Different training paradigms.** CRNCD adopts a two-stage training procedure that first trains a supervised model and then freezes it in the second stage. Contrarily, our framework adopts one-stage training in which the main and auxiliary branches help each other simultaneously.
- **Different distilled data.** While CRNCD distills all unlabeled data, our approach focuses on pseudo-base data. Here, pseudo-base refers to predictions belonging to the base classes within the main branch.
- **Different distillation weights.** CRNCD adopts a learnable weight function to control the distillation strength. We utilize the maximum auxiliary probability as an uncertainty-based weight, which provides a simpler yet effective mechanism for regulating distillation.

Besides the above statement, we conduct a thorough experiment to compare the different distillation strategies. Table 9 shows that distilling across all unlabeled data reduces novel-class performance, as novel data increases the likelihood of incorrect base-class prediction. Additionally, the learnable distillation weight in CRNCD performs poorly for GCD, causing a significant drop in performance. These results highlight the effectiveness of our proposed design.

Table 9: Comparison of different distillation strategies.

| Distillation strategy | CIFAR100 | | | CUB200 | | |
|---|---|---|---|---|---|---|
| | All | Base | Novel | All | Base | Novel |
| Distill on all unlabeled data | 81.8 | 83.8 | 77.8 | 66.4 | 75.4 | 61.8 |
| Learnable weight | 80.9 | 82.6 | 77.5 | 65.0 | 74.5 | 60.2 |
| Ours | **82.8** | 84.1 | **80.1** | **69.5** | 76.4 | **65.9** |

## A.8 Extended experiment on different regularizations

We have conducted the ablation study of different regularizations on three datasets in Table 4, and we here to include more results on the other datasets. As shown in Fig. 8, our proposed CDR achieves consistently great GCD performance, outperforming other regularization among all datasets. The result further demonstrates the superiority of our approach.

## A.9 Extended experiment on estimated category numbers.

Since the estimated category numbers from GCP Zhao et al. (2023) cover the partial datasets, we conduct comparison on other datasets under CMS estimation.

Table 10: Comparison of estimated category numbers on CIFAR100, FGVC-Aircraft, Herbarium-19 and Stanford Cars datasets.

| Methods | $K$ | CIFAR100 | | | FGVC-Aircraft | | | Herbarium-19 | | | Standford Cars | | |
|---|---|---|---|---|---|---|---|---|---|---|---|---|---|
| | | All | Base | Novel | All | Base | Novel | All | Base | Novel | All | Base | Novel |
| CMS Choi et al. (2024) | 97/98/666/152 | 79.6 | 83.2 | **72.3** | 55.2 | 60.6 | 52.4 | 37.4 | 56.5 | 27.1 | 51.7 | 68.9 | 43.4 |
| **Ours** | 97/98/666/152 | **80.0** | **84.3** | 71.4 | **69.4** | **78.5** | **64.8** | **46.0** | **62.3** | **37.2** | **56.6** | **73.7** | **48.4** |

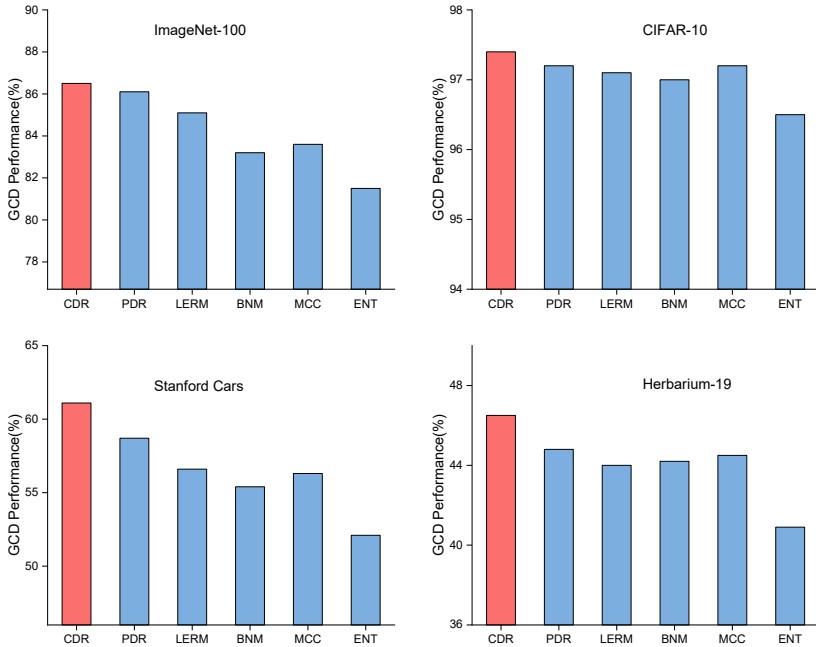

Figure 8: Comparison of different regularizations.

## A.10 RESULTS ON DINOv2

The proceeding experiments are conducted based on the pre-trained DINO Caron et al. (2021) model. To further validate the effectiveness of our method, we substitute the backbone with DI-NOv2 Oquab et al. (2023). As shown in Tables 11 and 12, the stronger backbone brings further improvement and our approach consistently outperforms other methods across all GCD datasets, with a notable 8.7% improvement on FGVC-Aircraft.

Table 11: GCD performance comparison on generic datasets utilizing DINOv2.

| Methods | CIFAR10 | | | CIFAR100 | | | ImageNet-100 | | |
|---|---|---|---|---|---|---|---|---|---|
| | All | Base | Novel | All | Base | Novel | All | Base | Novel |
| SimGCD Wen et al. (2023) | 98.8 | 96.9 | **99.7** | 88.5 | 89.3 | 86.9 | 88.5 | 96.2 | 84.6 |
| CiPR Hao et al. (2024) | 99.0 | 98.7 | 99.2 | 90.3 | 89.0 | **93.1** | 88.2 | 87.6 | 88.5 |
| SPTNet Wang et al. (2024) | - | - | - | - | - | - | 90.1 | 96.1 | 87.6 |
| Ours | **99.0** | 98.9 | 99.1 | **91.0** | 91.2 | 90.5 | **92.1** | 96.2 | **90.0** |

Table 12: GCD performance comparison on fine-grained datasets utilizing DINOv2.

| Distill strategy | CUB200 | | | Stanford Cars | | | FGVC-Aircraft | | | Herbarium-19 | | |
|---|---|---|---|---|---|---|---|---|---|---|---|---|
| | All | Base | Novel | All | Base | Novel | All | Base | Novel | All | Base | Novel |
| SimGCD Wen et al. (2023) | 74.9 | 78.5 | 73.1 | 71.3 | 81.6 | 66.4 | 63.9 | 69.9 | 60.9 | 58.7 | 63.8 | 56.2 |
| CiPR Hao et al. (2024) | 78.3 | 73.4 | **80.8** | 66.7 | 77.0 | 61.8 | - | - | - | 59.2 | 65.0 | **56.3** |
| SPTNet Wang et al. (2024) | 76.3 | 79.5 | 74.6 | - | - | - | - | - | - | -- | - | - |
| Ours | **78.7** | 79.5 | 78.3 | **79.5** | 91.8 | 73.5 | **72.6** | 77.3 | 70.3 | **60.2** | 71.9 | 54.0 |

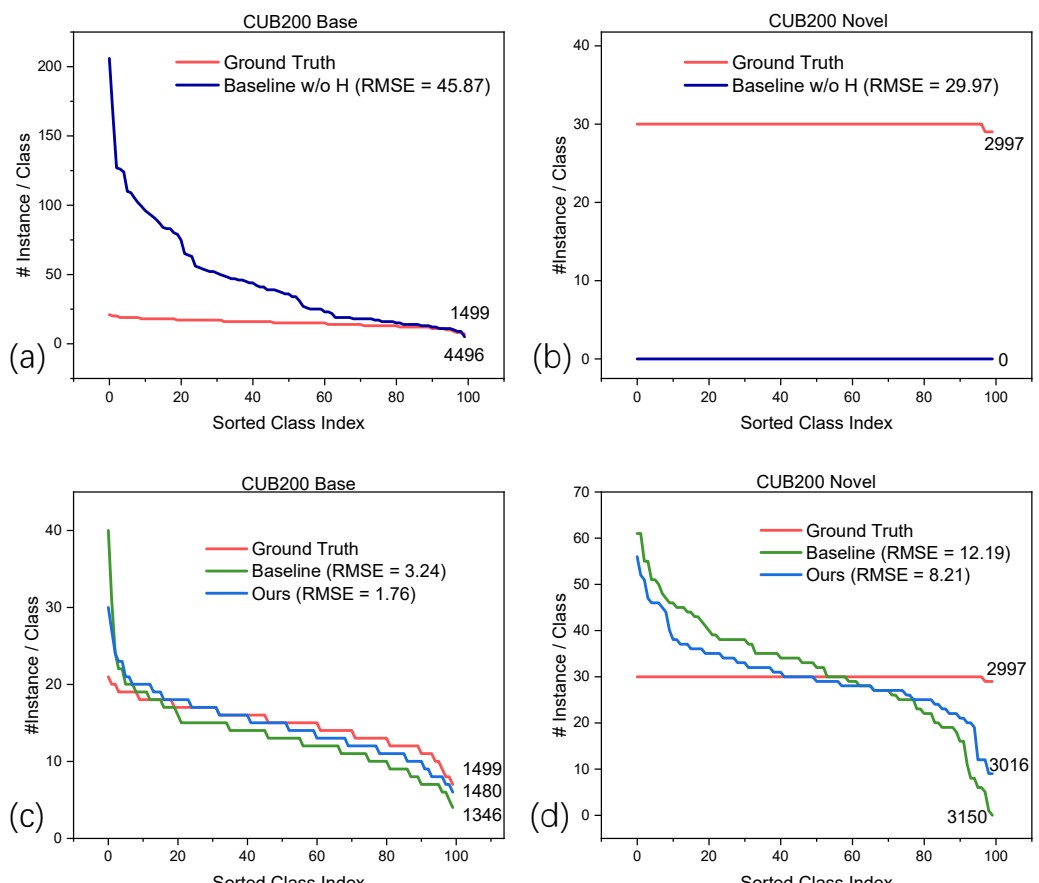

Figure 9: Class-wise prediction distributions on different methods. Concretely, Root Mean Squared Error (RMSE) is to measure the prediction distribution deviation with Ground Truth and the cumulative number is marked at the end of each curve.

## A.11   FURTHER ANALYSIS ON $H$

By definition, $H$ encourages more diverse predictions in the mini-batch. In fact, H plays an important role in balancing base and novel class performance in the parametric-based method. However, as discussed in Section 3.2, $H$ would hurt base class discrimination, resulting in degraded oracle base class accuracy. Here, we conduct a deep analysis of the effect of $H$ utilizing class-wise prediction distribution (CPD). Root Mean Squared Error(RMSE) is to quantify the class prediction distribution deviation with the Ground Truth.

As shown in Figure 9, we compare several methods, including a parametric-based baseline with and without $H$, as well as our proposed approach. When $H$ is removed, all samples are predicted as the base class, resulting in a significantly large RMSE for both base and novel CPDs, 45.87 and 29.97, respectively. From Figure 9(c) and (d), we observe that $H$ effectively refines the CPD, reducing the RMSE by 42.63 for base classes and 17.78 for novel classes. However, $H$ also introduces the side effect of misclassifying some base class samples. Specifically, the predicted base class samples are much lower than the ground truth, dropping from 1499 to 1346, which introduces noisy label learning during training. To mitigate this noisy learning, we propose a reciprocal learning framework, where the auxiliary branch provides more reliable pseudo labels to the main branch. Through cross-branch distillation, our method increases the number of predicted base class samples from 1346 to 1480. Furthermore, our CPD is closer to the ground truth, outperforming the baseline with RMSE reductions of 1.48 and 3.98 for the base and novel classes, respectively.

