# OpenReview forum: "Generalized Category Discovery  Utilizing Reciprocal Learning and Class-wise Distribution Regularization"
_ICLR.cc/2025/Conference — Submitted to ICLR 2025_

### Official Review · Reviewer_xWMZ · 2024-10-28

**Soundness:** 3
**Presentation:** 3
**Contribution:** 3
**Rating:** 6
**Confidence:** 4

**Summary:**

This paper proposes a Reciprocal Learning framework, introducing an auxiliary branch for base classification and class-wise distribution regularization to mitigate learning bias towards base classes. Extensive experiments on various datasets demonstrate the effectiveness and benefits of the proposed method.

**Strengths:**

The paper is easy to read and follow, and the results are convincing.

The proposed method is interesting, especially class-wise distribution regularization.

**Weaknesses:**

1. Could you provide an intuitive understanding of $m_k$?
2. It seems that the KL loss is similar to the approach by [1], which distills knowledge from a supervised model. Could you offer a more detailed analysis of the differences?
3. Experiments with DINOv2 are needed to better demonstrate the advantages of your methods.


[1] Gu et al. Class-relation Knowledge Distillation for Novel Class Discovery. ICCV2023.

**Questions:**

Please see weakness.

---

> ### Author Response · Authors · 2024-11-21
> **Response to Reviewer xWMZ (1/2)**
>
> **Q1:** Could you provide an intuitive understanding of $m_k$?
>
> **A1:** Thanks for your comment. By definition, $m_k$ represents the weighted average of mini-batch distributions, where the weight for each sample is its probability belonging to category $k$. Moreover, the $k$-th probability of $m_k$, denoted as $m(k, k)$, intuitively reflects the confidence that "the mini-batch contains at least one sample belonging to category $k$." The other probability of $m_k$ , $m(k,i),i \neq k$, reflects the affinity between category $i$ and category $k$.
>
> For instance, if the mini-batch does not contain any samples from class $k$, $m_k$ behaves like the average distribution of all samples. Conversely, if the mini-batch includes samples from class $k$, $m_k$ should be close to a one-hot distribution.
>
> ---
>
> **Q2:** It seems that the KL loss is similar to the approach by [1], which distills knowledge from a supervised model. Could you offer a more detailed analysis of the differences?
>
> **A2:** Thank you for highlighting this point. Although both our approach and CRNCD [1] adopt knowledge distillation, our method is fundamentally different from CRNCD. We first provide a detailed analysis of the key differences:
>
> - **Different tasks.** CRNCD [1] deals with novel class discovery (NCD) problems, where all unlabeled data belong to novel classes. In contrast, our focus is on GCD, where unlabeled data comprises both novel and base class samples. Applying CRNCD to GCD demonstrates unsatisfactory performance in GCD due to the intrinsic task difference, as shown in the following table.
>
> |           | CIFAR10  | CIFAR100 | ImageNet-100 | CUB      | Stanford cars | Aircraft | Herbarium-19 | Avg.     |
> | --------- | :--------: | :--------: | :------------: | :--------: | :-------------: | :--------: | :------------: | :--------: |
> | CRNCD [1] | 96.9     | 80.3     | 81.4         | 62.7     | 54.1          | 54.4     | 41.3         | 67.3     |
> | Ours      | **97.4** | **82.8** | **86.5**     | **69.5** | **61.1**      | **60.6** | **46.4**     | **72.0** |
>
> - **Different motivations for using distillation.** The distillation in CRNCD aims to improve **novel** class performance, whereas our distillation is intended to promote **base** class discrimination.
>
> - **Different training paradigms.** CRNCD adopts a **two-stage** training procedure that first trains a supervised model and then freezes it in the second stage. Contrarily, our framework is more efficient, which adopts **one-stage** training in which the main and auxiliary branches help each other simultaneously.
>
>
> - **Different distilled data.** While CRNCD employs distillation on **all unlabeled data**, our approach focuses on the **selective data** (i.e., pseudo-base data in our submission), which are predicted to the base classes in the main branch. The following additional experiments compare these two approaches and demonstrate that our method is more effective.
>
> | Distilled data       |   CUB200 |          |          | CIFAR100 |          |          |
> | -------------------- | --------:| -------- | -------- | --------:| -------- | -------- |
> |                      |      All | Base     | Novel    |      All | Base     | Novel    |
> | All unlabeled [1] |     66.4 | 75.4     | 61.8     |     81.8 | 83.8     | 77.8     |
> | Pseudo-base (ours)   | **69.5** | **76.4** | **65.9** | **82.8** | **84.1** | **80.1** |
>
>
> - **Different distillation weight assignments.** CRNCD adopts a **learnable weight** function to control the distillation strength. Instead, we utilize the maximum auxiliary probability as a **determined uncertainty-based weight**, which provides a **simpler and more effective** mechanism for regulating the distillation. This can be seen in the following experiments.
>
> | Weight type           |   CUB200 |          |          | CIFAR100 |          |          |
> | --------------------- | --------:| -------- | -------- | --------:| -------- | -------- |
> |                       |      All | Base     | Novel    |      All | Base     | Novel    |
> | Learnable [1]    |     65.0 | 74.5     | 60.2     |     80.9 | 82.6     | 77.5     |
> | Max. aux prob. (ours) | **69.5** | **76.4** | **65.9** | **82.8** | **84.1** | **80.1** |
>
>
>
> *[1] Gu P, Zhang C, Xu R, et al. Class-relation knowledge distillation for novel class discovery. ICCV2023.*

---

> ### Author Response · Authors · 2024-11-21
> **Response to Reviewer xWMZ (2/2)**
>
> **Q3:**  Experiments with DINOv2 are needed to better demonstrate the advantages of your methods.
>
> **A3:** Thank you for your valuable comment. Below, we present the GCD performance using the DINOv2 backbone on three representative datasets: CUB200, ImageNet-100, and Stanford Cars. As shown in the table below, our method consistently achieves the best overall performance. Particularly, our method obtains an 8.2\% improvement to SimGCD [1] on Stanford Cars dataset. Additional results on other datasets are provided in the revised submission, and our method consistently reaches the best performance.
>
> |                   |   CUB200 |          |          | ImagNet-100 |          |          | Stanford Cars |          |          |
> | ----------------- | --------:| -------- | -------- | -----------:| -------- | -------- | -------------:| :--------: | -------- |
> |                   |      All | Base     | Novel    |         All | Base     | Novel    |           All | Base     | Novel    |
> | SimGCD-DINOv2 [1] |     74.9 | 78.5     | 73.1     |        88.5 | 96.2     | 84.6     |          71.3 | 81.6     | 66.4     |
> | SPTNet-DINOv2 [2] |     76.3 | 79.5     | 74.6     |        90.1 | 96.1     | 87.6     |             - |    -      | -        |
> | CiPR-DINOv2 [3]   |     78.3 | 73.4     | **80.8** |        88.2 | 87.6     | 88.5     |          66.7 | 77.0     | 61.8     |
> | Ours-DINOv2       | **78.7** | **79.5** | 78.3     |    **92.1** | **96.2** | **90.0** |      **79.5** | **91.8** | **73.5** |
>
> *[1] Wen X, Zhao B, Qi X. Parametric classification for generalized category discovery: A baseline study. ICCV2023.\
> [2] Wang H, Vaze S, Han K. SPTNet: An efficient alternative framework for generalized category discovery with spatial prompt tuning. ICLR2024.\
> [3] Hao S, Han K, Wong K Y K. Cipr: An efficient framework with cross-instance positive relations for generalized category discovery. TMLR2024.*

---

> > ### Comment · Reviewer_xWMZ · 2024-11-27
> >
> > Thank you for your detailed feedback. I truly appreciate your thoughtful analysis.
> >
> > Regarding the distinction with CRNCD, I respectfully disagree with the suggestion that there are "different tasks and different motivations for using distillation." The difference between NCD and GCD is minimal, and, as shown in Table 3, CRNCD also yields improvements on known classes.
> >
> > Additionally, I recognize the simplicity and efficiency of the one-stage design. Based on these considerations, I have decided to maintain my initial score.

---

> > > ### Author Response · Authors · 2024-11-27
> > > **Further Discussion on CRNCD**
> > >
> > > We sincerely thank you for recognizing the novelty of our one-stage distillation framework and for your thoughtful and constructive comments regarding the distinction between CRNCD [1] and our approach. We greatly value your feedback and would like to provide additional discussion to clarify this point further.
> > >
> > > First, CRNCD has a clear and distinct motivation centered on improving novel class recognition. As stated in its abstract, CRNCD introduces "*a novel knowledge distillation framework, which utilizes our class-relation representation to regularize the learning of novel classes*." In contrast, we reveal the inferior base class discrimination within the parametric-based GCD methods, which **motivates our cross-branch distillation approach** to enhance base class performance.
> > > We kindly remind you that improving novel class recognition does not necessarily imply better base class discrimination, and vice versa.
> > >
> > > Second, while CRNCD improves known class accuracy, it does so indirectly through better representation learning. CRNCD achieves a 1–2% accuracy improvement on known classes, as explained in Section 4.2 of CRNCD: *“This, in turn, enhances representation learning and indirectly improves the performance of known classes*.” This statement highlights that the improvement in known (base) class accuracy is the **indirect outcome** of distillation. Unlike CRNCD, our approach focuses on pseudo-base data distillation, which **directly strengthens** base class performance.
> > >
> > > Third, **CRNCD encounters challenges in the GCD task due to task differences.** When applied to the GCD task, CRNCD demonstrates suboptimal performance, particularly for novel classes.   Since unlabeled data **involves base class samples,** the distillation in CRNCD induces **learning bias toward base classes.**  As a result, CRNCD performs better on base classes but exhibits **poor performance on novel classes**. As shown in the table below, it lags behind SimGCD [2] by 6.3% in novel accuracy on CIFAR100. This highlights the task difference between GCD and NCD. In contrast, our distillation approach is fundamentally distinct, incorporating a **one-stage distillation framework**, **pseudo-base distilled data**, and **uncertainty-based distillation weight**. These designed elements make our approach more effective for the GCD task.
> > >
> > > |            | CIFAR100 |          |          | CUB200   |          |          | Stanford Cars |          |          |
> > > | ---------- | --------: | -------- | -------- | --------: | -------- | -------- | -------------: | -------- | -------- |
> > > |            | All      | Base     | Novel    | All      | Base     | Novel    | All           | Base     | Novel    |
> > > | SimGCD [2] | 80.1     | 81.2     | 77.8     | 60.1     | 65.6     | 57.7     | 53.8          | 71.9     | 45.0     |
> > > | CRNCD [1]  | 80.3     | **84.7** | 71.5     | 62.7     | 71.6     | 58.2     | 54.1          | 75.7     | 43.7     |
> > > | Ours       | **82.8** | 84.1     | **80.1** | **69.5** | **76.4** | **65.9** | **61.1**      | **76.6** | **53.6** |
> > >
> > >
> > >
> > > Overall, this comparison underscores how our method has been specifically designed to address the unique challenges of GCD, resulting in a more balanced and effective solution for this task. If you have any further concerns, we are more than happy to discuss them with you.
> > >
> > >
> > > *[1] Gu P, Zhang C, Xu R, et al. Class-relation knowledge distillation for novel class discovery. ICCV2023.\
> > > [2] Wen X, Zhao B, Qi X. Parametric classification for generalized category discovery: A baseline study. ICCV2023.*

---

> > > > ### Comment · Reviewer_xWMZ · 2024-11-27
> > > >
> > > > Thanks for the additional comment, which addresses most of my concerns.
> > > >
> > > > Can you provide more details on how you implement crNCD? Can you apply the naive distillation loss (w/o reweighting) of crNCD to the SimGCD? Considering the training cost, results on fine-grained datasets are acceptable.
> > > >
> > > > Note that, given the limited time, the additional results are not essential.

---

> ### Author Response · Authors · 2024-11-28
> **More Implementations**
>
> Thank you for your valuable comments. We are glad to have addressed most of your concerns and would like to provide further discussion on CRNCD.
>
> **Q1:** Can you provide more details on how you implement crNCD? \
> **A1:** CRNCD [1] builds upon the UNO [2] framework, which employs optimal transport to assign labels. However, as pointed out by SimGCD [3], **this framework may not generate reliable pseudo-labels** in the GCD task. To ensure a fair comparison, we incorporate the distillation function [4] from CRNCD into our parametric baseline. Specifically, we follow a two-step procedure:
> 1. We first train a supervised model using the supervised loss function  $L_{sup}$, where the training data consists solely of labeled samples.
> 2. We then freeze the supervised model and proceed to train the parametric-based model with a combination of supervised loss, self-supervised loss, and distillation loss:\
> $L$ = $\lambda$ $L_{sup}$ + ( 1- $\lambda$)($L_{self}$ - $\epsilon H$） +  $\alpha$  $L_{rkd}$, \
> where $L_{rkd}$  represents the distillation loss from CRNCD, $\lambda,\epsilon$ are common hyper-parameters in the parametric-based methods, and $\alpha$ controls the strength of the distillation. The distillation is performed on all unlabeled data, and we follow the default distillation hyper-parameters used in CRNCD for consistency.
>
> ---
>
> **Q2:**  Can you apply the naive distillation loss (w/o reweighting) of crNCD to the SimGCD?\
> **A2:** We have applied the naive distillation loss (without reweighting) from CRNCD to SimGCD. To further investigate the effect of distillation, we experiment with two different values for the distillation strength: $\alpha = 0.1$(the default value in CRNCD), and $\alpha = 0.5$. Additionally, we provide results of our distillation method without the CDR loss for comparison. The performance on fine-grained datasets is summarized in the table below, where KD refers to the naive distillation approach.
>
>
>
> |                            |   CUB200 |          |          | Stanford Cars |          |          | Aircraft |          |          |
> | -------------------------- | --------:| -------- | -------- | -------------:| -------- | -------- | --------:| -------- | -------- |
> |                            |      All | Base     | Novel    |           All | Base     | Novel    |      All | Base     | Nove     |
> | SimGCD [3]                     |     60.3 | 65.6     | 57.7     |          53.8 | 71.9     | 45.0     |     54.2 | 59.1     | 51.8     |
> | SimGCD+KD ($\alpha = 0.1$) |     63.0 | 74.7     | 57.1     |          56.1 | 78.3     | 45.4     |     53.4 | 60.3     | 50.0     |
> | SimGCD+KD ($\alpha = 0.5$) |     63.6 | **76.9** | 57.0     |          57.0 | **80.0** | 45.9     |     54.4 | **63.0** | 50.1     |
> | Ours w/o CDR               | **66.5** | 75.9     | **61.8** |      **57.9** | 75.8     | **49.3** | **58.2** | 62.0     | **56.3** |
>
> The results indicate that naive distillation can improve SimGCD performance on the CUB200 and Stanford Cars datasets, with similar performance on Aircraft. Notably, the improvement from KD primarily comes from the base class, with stronger distillation ($\alpha$) leading to **higher base class accuracy**. However, KD shows **limited impact on novel class performance**, resulting in degradation on CUB200 and Aircraft.  The unbalanced performance suggests a learning bias in the naive distillation. In contrast, our distillation method achieves the best results, with **consistent improvements across both base and novel classes**.
>
> *[1] Gu P, Zhang C, Xu R, et al. Class-relation knowledge distillation for novel class discovery. ICCV2023.\
> [2] Fini E, Sangineto E, Lathuilière S, et al. A unified objective for novel class discovery. ICCV2021.\
> [3] Wen X, Zhao B, Qi X. Parametric classification for generalized category discovery: A baseline study. ICCV2023.\
> [4] https://github.com/kleinzcy/Cr-KD-NCD/blob/main/model/losses.py#L38*

---

> > ### Comment · Reviewer_xWMZ · 2024-11-28
> >
> > Thanks a lot for your detailed feedback. All my concerns are resolved. Good luck.

---

> > > ### Author Response · Authors · 2024-11-28
> > > **Thanks for the feedback**
> > >
> > > We are pleased to have addressed all of your concerns. Your constructive comments have significantly contributed to improving the quality of our work. Thank you once again for your thoughtful review！

---

### Official Review · Reviewer_UvsP · 2024-10-31

**Soundness:** 3
**Presentation:** 3
**Contribution:** 3
**Rating:** 6
**Confidence:** 5

**Summary:**

This paper introduces a novel approach to address the challenge of Generalized Category Discovery (GCD) by utilizing Reciprocal Learning (RLF) and Class-wise Distribution Regularization (CDR). In RLF, the authors address the issue of poor base discrimination in current methods by introducing a base-only classifier and distilling the distributions between this classifier and the main classifier on filtered pseudo-base data. In CDR, the authors enhance novel class performance by maximizing the similarity between class-wise expected distributions from two views. Experimental results highlight the superiority of this method over existing state-of-the-art techniques in both effectiveness and generalizability.

**Strengths:**

1. The motivation behind the Reciprocal Learning framework is clearly articulated, and the solution is both simple and efficient.
2. The design of class-wise distribution regularization is elegant, and its improvements are impressive.
3. The method achieves significant performance gains, demonstrating its effectiveness.
4. The paper is well-written and easy to follow.

**Weaknesses:**

1. While class-wise distribution regularization serves as an effective regularization term, its motivation is unclear. The authors claim it primarily addresses the learning bias towards base classes caused by RLF and sharpens model predictions. However, CDR is applied to both branches, including the base classifier branch where it affects only the pseudo base classes. This could potentially reinforce the bias towards base classes and is contradictory to the stated motivation. Further clarification and an ablation study focusing on the impact of CDR in each branch would be beneficial. For instance, it would be useful to compare the effects of applying CDR exclusively to the main branch versus its application across both branches.
2. The reciprocal learning framework bears a resemblance to the approach presented in [1]. In that work, an additional base classifier head is trained with labeled base class data, and the distribution between this base-class classifier and the main classifier is distilled with a weight function (Eq. 8 in [1]) to reduce the impact of noisy unlabeled data learning. While there are differences in details, such as the design of the weight function (specifically, $\max(\mathbf{p}_{b, i}^{aux})$ in Eq. 5 of this paper versus Eq. 8 in [1]) and the training method of the base classifier, the framework and the final loss function (Eq. 5 in this paper versus Eq. 9 in [1]) are quite similar. A thorough analysis of the necessity of these modifications and a discussion of the novelty claim would strengthen the contribution.

[1] Peiyan Gu, Chuyu Zhang, Ruijie Xu, and Xuming He. Class-relation knowledge distillation for
novel class discovery.

**Questions:**

Main questions and suggestions:

Please refer to the Weaknesses section for detailed questions and suggestions.

Minor questions and suggestions:
1. I noticed that the contrastive learning loss is applied only to labeled data (Eq. 3). Could you explain the reason for excluding it from unlabeled data?
2. In Eq. 5, is the item $\max\(\mathbf{p}_{b,i}^{aux}\)$  detached during the computation?
3. The method as a whole lacks a distinctive name. It would be beneficial for clarity and reference if the authors could provide a name for their approach.

---

> ### Author Response · Authors · 2024-11-21
> **Response to Reviewer UvsP (1/3)**
>
> **Q1：** While class-wise distribution regularization serves as an effective regularization term, its motivation is unclear. The authors claim it primarily addresses the learning bias towards base classes caused by RLF and sharpens model predictions. However, CDR is applied to both branches, including the base classifier branch where it affects only the pseudo base classes. This could potentially reinforce the bias towards base classes and is contradictory to the stated motivation. Further clarification and an ablation study focusing on the impact of CDR in each branch would be beneficial. For instance, it would be useful to compare the effects of applying CDR exclusively to the main branch versus its application across both branches.
>
>
> **A1：** Thank you for your insightful comment. According to our Theorem 2, CDR aims to implicitly increase prediction confidence. Since **each class is treated equally**, through expected class-wise distribution, CDR effectively mitigates the base class learning bias. When applied to the auxiliary branch, CDR also benefits base class learning with minimal impact on the novel class performance of the main branch.
> To further validate the role of CDR, we conduct an ablation study comparing: w/o CDR, CDR on the main branch, and CDR on both branches.
>
>
>
>
>
> |             |      CUB200 |      |       |      |   CIFAR100 |      |       |      |
> | ----------- | -----------:| ---- | ----- | ---- | ----------:| ---- | ----- | ---- |
> |             |         All | Base | Novel | OB   |        All | Base | Novel | OB   |
> | w/o CDR     |        66.5 | 75.9 | 61.8  | 87.2 |       80.8 | 83.8 | 74.7  | 86.2 |
> | CDR on main | 68.7 (↑2.2) | 75.3 | 65.4  | 86.7 | 82.3 (↑1.5) | 83.5 | 80.0  | 85.6 |
> | CDR on both | 69.5 (↑↑0.8) | 76.4 | 65.9  | 87.3 |  82.8(↑↑0.5) | 84.1 | 80.1  | 86.0 |
>
>
> - From the above table, applying CDR exclusively to the main branch effectively mitigates the learning bias and boosts novel class performance.
> -  When applied to both branches, CDR maintains novel class accuracy on the main branch while slightly improving base class performance. **Since the AUX and CLS features exist in separate spaces**, applying CDR additionally to the AUX branch does not exacerbate the learning bias toward base classes.
>
> The clarification and ablation study have been included in our revised submission.

---

> ### Author Response · Authors · 2024-11-21
> **Response to Reviewer UvsP (2/3)**
>
> *Q2:** The reciprocal learning framework bears a resemblance to the approach presented in [1]. In that work, an additional base classifier head is trained with labeled base class data, and the distribution between this base-class classifier and the main classifier is distilled with a weight function (Eq. 8 in [1]) to reduce the impact of noisy unlabeled data learning. While there are differences in details, such as the design of the weight function (specifically,  Eq. 5 of this paper versus Eq. 8 in [1]) and the training method of the base classifier, the framework and the final loss function (Eq. 5 in this paper versus Eq. 9 in [1]) are quite similar. A thorough analysis of the necessity of these modifications and a discussion of the novelty claim would strengthen the contribution.
>
> **A2:** Thanks for your thoughtful comment. We would clarify the fundamental difference between CRNCD [1] and our approach. Besides, Comparative experiments validate the necessity of our design.
>
> - **Different motivations for using distillation.** The distillation in CRNCD aims to improve **novel** class performance, whereas our distillation is intended to promote **base** class discrimination.
>
> - **Different training paradigms.** CRNCD adopts a **two-stage** training procedure that first trains a supervised model and then freezes it in the second stage. Contrarily, our framework utilizes an AUX token to enable **one-stage** training in which the main and auxiliary branches help each other simultaneously.  Without the AUX token, the base classifier and full classifier are forced to share the same feature space. This limits the effectiveness of distillation, resulting in a significant performance drop, as shown in the table below.
>
> |              | CUB200   |          |          | CIFAR100 |          |          |
> | ------------ | --------: | -------- | -------- | --------:| -------- | -------- |
> |              | All      | Base     | Novel    |      All | Base     | Novel    |
> | Ours w/o AUX | 65.9     | 73.8     | 61.9     |     81.0 | 81.9     | 79.2     |
> | Ours         | **69.5** | **76.4** | **65.9** | **82.8** | **84.1** | **80.1** |
>
>
>
>
> - **Different distilled data.** While CRNCD employs distillation on **all unlabeled data**, our approach focuses on the **selective data** (i.e., pseudo-base data in our submission), which are predicted to the base classes in the main branch. The following additional experiments compare these two approaches and demonstrate that our method is more effective.
>
> | Distilled data     |   CUB200 |          |          | CIFAR100 |          |          |
> | ------------------ | --------:| -------- | -------- | --------:| -------- | -------- |
> |                    |      All | Base     | Novel    |      All | Base     | Novel    |
> | All unlabeled[1]    |     66.4 | 75.4     | 61.8     |     81.8 | 83.8     | 77.8     |
> | Pseudo-base (ours) | **69.5** | **76.4** | **65.9** | **82.8** | **84.1** | **80.1** |
>
>
> - **Different distillation weight assignments.** CRNCD adopts a **learnable weight** function to control the distillation strength. Instead, we utilize the maximum auxiliary probability as a **determined uncertainty-based weight**, which provides a **simpler and more effective** mechanism for regulating the distillation. This can be seen in the following experiments.
>
> | Weight type           |   CUB200 |          |          | CIFAR100 |          |          |
> | --------------------- | --------:| -------- | -------- | --------:| -------- | -------- |
> |                       |      All | Base     | Novel    |      All | Base     | Novel    |
> | Learnable[1]          |     65.0 | 74.5     | 60.2     |     80.9 | 82.6     | 77.5     |
> | Max. aux prob. (ours) | **69.5** | **76.4** | **65.9** | **82.8** | **84.1** | **80.1** |
>
>
>
>
>
>
> *[1] Gu P, Zhang C, Xu R, et al. Class-relation knowledge distillation for novel class discovery. ICCV2023.*

---

> ### Author Response · Authors · 2024-11-21
> **Response to Reviewer UvsP (3/3)**
>
> **Q3：** I noticed that the contrastive learning loss is applied only to labeled data (Eq. 3). Could you explain the reason for excluding it from unlabeled data?
>
> **A3：** Thanks for your insightful comment. We apply contrastive learning on only labeled samples for two reasons.
> - We found the official implementation of SimGCD applies InfoNCE to all samples in the mini-batch. However, InfoNCE would also push apart same-class features which contradicts the objective of SupCon loss and impairs discrimination.
> - By removing self-CL, we observed an improvement in oracle base accuracy, increasing from 83.6% to 84.4% on CIFAR-100.
>
> To make this point clearer, we have included the explanation above in the revised submission.
>
> ---
>
> **Q4:** In Eq. 5, is the item $\max \left(\mathbf{p}_{b, i}^{a u x}\right)$ detached during the computation?
>
> **A4:** Yes. Specifically, our ${L}\_{KL}$ would first freeze the auxiliary probability distribution, ensuring that the gradients do not flow back to the AUX branch. $\max \left(\mathbf{p}_{b, i}^{a u x}\right)$ is naturally detached in the operation. To ensure clarity, we have included this implementation detail in the revision.
>
> ---
> **Q5:** The method as a whole lacks a distinctive name. It would be beneficial for clarity and reference if the authors could provide a name for their approach.
>
> **A5:** Thanks for your advice, we prefer to name our method as "RLCD", derived from "**R**eciprocal **L**earning and **C**lass-wise **D**istribution regularization".

---

> > ### Comment · Reviewer_UvsP · 2024-11-25
> >
> > Thank you for your detailed feedback. I appreciate the time and effort you put into addressing my concerns. This has resolved most of my issues, so I will maintain my score.

---

> > > ### Author Response · Authors · 2024-11-25
> > > **Thanks for the feedback**
> > >
> > > Thank you once again for your valuable time and effort in reviewing our work and considering our rebuttal. Your thoughtful insights have been instrumental in helping us enhance the quality and clarity of our paper.
> > >
> > > In our rebuttal, we have emphasized the fundamental differences between our method and CRNCD, as well as conducted extensive experiments to clarify the motivation behind class-wise distribution regularization (CDR). We are encouraged to hear that most of your concerns have been addressed.
> > >
> > > With this in mind, **we kindly and humbly request your reconsideration of the scoring**. We believe that our one-stage reciprocal learning framework and the effective CDR loss introduced in our work make meaningful contributions to the field and offer valuable insights that could benefit the ICLR community.
> > >
> > > Thank you again for your time and thoughtful consideration.

---

### Official Review · Reviewer_hpPJ · 2024-11-02

**Soundness:** 3
**Presentation:** 3
**Contribution:** 2
**Rating:** 5
**Confidence:** 4

**Summary:**

The paper introduces a novel method called RLF to improve the reliability of pseudo-labels in Generalized Category Discovery (GCD) tasks. RLF incorporates an AUX token that learns base categories to enhance pseudo-label quality, along with a CDR module to regularize novel samples. Extensive experiments and ablation studies across multiple datasets demonstrate the effectiveness of this approach, showing superior performance in GCD tasks.

**Strengths:**

1.	This article is well-written and well-organized.
2.	The paper addresses the often-overlooked bias in pseudo-labeling through the proposed AUX token and regularization.
3.	The proposed method significantly improves performance and provides more reliable pseudo-labels for classifier learning.
4.	The paper includes comprehensive quantitative and qualitative visualizations to demonstrate the effectiveness of the method.

**Weaknesses:**

1.	Missing Relevant Works: The paper overlooks key recent works like CMS and InfoSieve.
2.	Missing Self-Contrastive Learning (CL): SimGCD also uses self-CL in representation learning, but this is absent in Section 3.1.
3.	Lacking Deep Reason Analysis: The authors aim to reduce bias in pseudo-labels but don't analyze the root causes deeply, even though they explain the motivation in Section 3.2-whether it's due to weak representations, class maximum regularization in SimGCD, or something else.
4.	Confusing Sentence: What is the meaning of K in line 234: ‘K is the number of base and novel classes, while K denotes the number of base classes in the auxiliary branch’

**Questions:**

See weaknesses 3 and 4.

---

> ### Author Response · Authors · 2024-11-21
> **Response to Reviewer hpPJ (1/2)**
>
> **Q1:** Missing Relevant Works: The paper overlooks key recent works like CMS and InfoSieve.
>
> **A1:** Thank you for pointing this out. We have now incorporated CMS [1] and InfoSieve [2] into the related works section and included their results in Tables 1 and 2 of the revised manuscript. The results show that **our method consistently outperforms both CMS and InfoSieve across all seven datasets**.
>
> It's worth noting that InfoSieve achieves close performance to our approach on CUB200. Actually, InfoSieve fine-tunes the last two blocks of the backbone for fine-grained datasets, whereas our approach only fine-tunes the last block. **When we adopt the same fine-tuning paradigm as InfoSieve, our method achieves further improvements.**
>
>
> | CUB200                          | All      | Base     | Novel    |
> | ------------------------------- | -------- | -------- | -------- |
> | InfoSieve [2] (last two blocks) | 69.4     | 77.9     | 65.2     |
> | Ours (last block)               | 69.5     | 76.4     | 65.9     |
> | Ours (last two blocks)          | **71.9** | **79.5** | **68.1** |
>
> *[1] Choi S, Kang D, Cho M. Contrastive mean-shift learning for generalized category discovery. CVPR2024.\
> [2] Rastegar S, Doughty H, Snoek C. Learn to categorize or categorize to learn? self-coding for generalized category discovery. NeurIPS2023.*
>
> ---
>
> **Q2:** Missing Self-Contrastive Learning (CL): SimGCD also uses self-CL in representation learning.
>
> **A2:** Thank you for your detailed observation. While it is true that SimGCD incorporates self-CL in representation learning, we would like to clarify that **our method does not utilize self-CL**.
>
> This design was made because the InfoNCE loss in SimGCD tends to push same-class features apart, which can harm the discrimination of the base classes. **By removing self-CL, we observed an improvement in oracle base accuracy**, increasing from 83.6% to 84.4% on CIFAR-100. To make this point clearer, we have included the explanation above in the revised submission.

---

> ### Author Response · Authors · 2024-11-21
> **Response to Reviewer hpPJ (2/2)**
>
> **Q3:** Lacking Deep Reason Analysis: The authors aim to reduce bias in pseudo-labels but don't analyze the root causes deeply, even though they explain the motivation in Section 3.2-whether it's due to weak representations, class maximum regularization in SimGCD, or something else.
>
> **A3:** Thanks for your constructive suggestion. To address this concern, we have conducted a **step-by-step loss analysis** of the parametric baseline on CUB200. The baseline consists of three loss components: supervised learning $L_{\text{sup}}$, self-supervised learning $L_{\text{self}}$, and mean entropy regularization $H$.  Moreover, we introduce three metrics:
> - **OB**: oracle base class accuracy to evaluate pseudo-label quality.
> - **N2N**: represents the percentage of novel class samples correctly classified as novel.
> - **B2B**: indicates the percentage of base class samples correctly classified as base.
>
> The results are summarized in the table below:
>
> | $L_{\text{sup}}$ | $L_{\text{self}}$ | $H$ | OB (Acc.)      | N2N (%) | B2B (%) |
> | ----------------- | ------------------ | ---- | ---------- | ------ | ------ |
> | ✓                 |                    |      | 86.7       | 0      | 100    |
> | ✓                 | ✓                  |      | 85.6(↓1.1) | 0      | 100    |
> | ✓                 | ✓                  | ✓    | 83.9(↓↓1.7) | 93.1   | 69.8   |
>
>
>
> - Firstly,  ${L}_{\text{sup}}$ builds a strong discriminative model for base classes as only the base training data is correctly labeled. Since novel classes are unlabeled initially with prototypes randomly initialized, **all unlabeled data will be classified as the base**, thus leading to 0% N2N and 100% B2B.
>
> - Additionally, adding ${L}\_{\text{self}}$ still results in all unlabeled novel data being misclassified. Due to the absence of annotations for novel data, the novel class probability is rather low. Moreover, ${L}_{\text{self}}$ adopts a shrink-soft probability distillation mechanism, which essentially **increases the incorrect confidence of novel data**. Training with noisy labels on novel samples degrades the model’s representation, causing a decline in OB (86.7->85.6) performance.
>
> - Furthermore, incorporating $H$ encourages the model to make more uniform predictions across classes, resulting in a **higher proportion of unlabeled data being classified as novel**. This leads to an increase in N2N (0->93.1) but a decrease in B2B (100->69.8). Furthermore, the self-distillation of $L_{\text{self}}$ would amplify the impact of these misclassifications, exacerbating noisy label learning and further degrading OB (85.6->83.9) performance.
>
> - To summarize, the above results indicate that **both $L_{\text{self}}$ and $H$ contribute to unreliably pseudo-labels**. This motivates us to introduce a more robust base-class expert to mitigate the noisy base labels and enhance discrimination.
>
>
> The above discussion is included in the revised submission.
>
> ---
> **Q4:** Confusing Sentence: What is the meaning of K in line 234: ‘K is the number of base and novel classes, while K denotes the number of base classes in the auxiliary branch’
>
> **A4:** Thank you for pointing out this. To clarify, the CDR is applied to both branches. In the main branch, $K$ represents the total number of classes ( base + novel). While in the auxiliary branch, $K$ specifically denotes the number of base classes. To avoid further confusion, we have revised the paper to distinguish between these two meanings by using $K^{main}$ and $K^{aux}$ in the updated version.

---

> > ### Comment · Reviewer_hpPJ · 2024-11-25
> >
> > Thank you for your detailed response, which has addressed most of my concerns. However, I still have one issue I’d like to discuss further.
> >
> > From the table in Question 3 and your response, it is clear that H is the primary cause of inaccurate pseudo-labels. However, in your reply to Question 3, you seem to focus more on designing a robust classifier to optimize \( L_{\text{self}} \), rather than addressing the root cause of the problem (H) directly.
> >
> > I hope you can conduct further research and analysis on H. Once again, thank you for your thoughtful response, and I will maintain my original score.

---

> ### Author Response · Authors · 2024-11-26
> **Further research and analysis on H**
>
> Thank you once again for your valuable time and effort in reviewing our work and considering our rebuttal. We are glad to address your main concerns and remain open to further discussion regarding $H$.
>
> First, we would like to clarify the relationship between $H$ and $L_{self}$.
> - **$H$ is dependent on $L_{self}$.** While $H$ encourages more predicted novel class samples, its impact is minimal when $L_{self}$ is removed. As shown in the table below, without $L_{self}$, the model suffers a significant N2N drop from 93.1% to 18.8%. $L_{self}$ essentially increases prediction confidence, which facilitates novel class prototype learning and attracts more novel class samples.  In this way, our approach refines the pseudo-label quality, which **directly enhances** $L_{self}$  and **indirectly influences $H$**.
>
>
>
> | Description             | +$L_{{sup}}$ | +$L_{{self}}$ | +$H$ | OB (Acc.)  | N2N (%) | B2B (%) |
> | ----------------------- | ------------ | ------------- | ---- | ---------- | ------- | ------- |
> | Supervised-only         | ✓            |               |      | 86.7       | 0       | 100     |
> | Baseline w/o $H$        | ✓            | ✓             |      | 85.6(↓1.1) | 0       | 100     |
> | Baseline w/o $L_{self}$ | ✓            |               | ✓    | 84.4(↓2.3) | 18.8    | 98.1    |
> | Baseline                | ✓            | ✓             | ✓    | 83.9(↓2.8) | 93.1    | 69.8    |
>
>
> To further investigate the effect of $H$, we introduce class-wise prediction distribution (CPD) and show the result on CUB200. CPD provides a detailed view of how the model assigns predictions to each class. In particular, we use the Root Mean Squared Error (RMSE) to quantify the deviation between the CPD and the ground truth distribution, with smaller RMSE values indicating better results. The results are summarized in the table below:
>
> |                  | RMSE-Base(↓) | RMSE-Novel(↓) | Total Base Number | Total Novel Number |
> | ---------------- | ------------ | ------------- | ----------------- | ------------------ |
> | Ground truth     | 0            | 0             | 1499              | 2997               |
> | Baseline w/o $H$ | 45.87        | 29.97         | 4496              | 0                  |
> | Baseline         | 3.24         | 12.19         | 1346              | 3150               |
> | Ours             | **1.76**     | **8.21**      | **1480**          | **3016**           |
> - **$H$ steers CPD toward the ground truth.** When $H$ is removed, all unlabeled data are classified as base classes, resulting in a large RMSE in both base and novel CPDs: 45.87 and 29.97, respectively. Incorporating $H$ encourages more diverse predictions and **induces more predicted novel samples**. Therefore, $H$ effectively reduces the RMSE: 45.87 -> 3.24 for base classes, and  29.97 -> 12.19 for novel classes.
>
> - **Our approach compensates $H$ for better base performance.** Actually, $H$ leads to misclassification of some base class samples, and the total predicted number of base samples is lower than the ground truth (1346 vs. 1499). To mitigate this, our method introduces cross-branch distillation during training, which helps reduce the misclassification. As a result, our predicted base number is closer to the ground truth number (1480 vs. 1499). Moreover, our method achieves a **lower RMSE** than the baseline:  3.24 -> 1.76 for base classes, and 12.19 -> 8.21 for novel classes.
>
>
> While $H$ may hinder base class discrimination, it provides a simple and efficient way to improve novel class recognition. Therefore, our motivation is to mitigate the negative impact of $H$ and leverage its strengths to enhance novel class recognition.
> This detailed discussion and the experiments have been included in the revised submission (see Appendix A.11). Once again, thanks for your insightful suggestion.

---

> > ### Comment · Reviewer_hpPJ · 2024-11-27
> >
> > I appreciate your detailed response. While \( L_{self} \) is influenced by \( H \), it degrades the quality of the pseudo-labels, a point not adequately reflected in the motivation or manuscript. Therefore, I will maintain the original grade. Additionally, there is an error in the book quotation in line 272 of the latest article.

---

> ### Author Response · Authors · 2024-11-27
> **Further Clarification**
>
> Thanks for your thoughtful comment. We have corrected the book quotation and reviewed the manuscript to prevent similar issues.  Besides, we have revised the loss analysis in Table R1 to include the influence between $\mathcal{L}_{\text{self}}$ and $H$ in Lines 227-232 as follows：
>
> > The combination of $\mathcal{L}\_{\text{sup}}$ and $H$ increases novel class prediction, slightly improving N2N but causing a decline in OB.  This decline occurs because $H$ encourages a balanced average prediction.  When all three losses are applied, the model predicts more novel class samples, leading to 93.1\% N2N. This is because $\mathcal{L}\_{\text{self}}$ increases the novel class confidence, facilitating novel class prototype learning. However, $H$ simultaneously drives the misclassification of some base samples, while $\mathcal{L}\_{\text{self}}$ amplifies incorrect novel confidence, further deteriorating the pseudo-label quality. These findings suggest that both $\mathcal{L}\_{\text{self}}$ and $H$ contribute to unreliable pseudo-labels.
>
>
> Furthermore, we have modified our motivation in Lines 233–235 as follows:
> > Besides, $\mathcal{L}\_{\text{self}}$ is particularly  susceptive to $H$ and exacerbates noisy learning.  This motivates us to introduce a more robust base-class expert to assist $\mathcal{L}\_{\text{self}}$. By providing more reliable base labels, this intervention aims to enhance discrimination and mitigate the side effects of $H$.
>
>
> The revised version has been updated. We sincerely appreciate your valuable feedback, which has helped us improve the clarity and rigor of our work.

---

> > ### Author Response · Authors · 2024-12-02
> > **Appreciating Your Reviews and Humbly Asking for Feedback**
> >
> > Dear Reviewer hpPJ,
> >
> > We sincerely appreciate your valuable time devoted to reviewing our manuscript. We would like to gently remind you of the **approaching deadline** for the discussion phase.
> >
> > In our latest revision, we have included the point that $\mathcal{L}_{\text{self}}$ degrades the pseudo-label quality with the influence of $H$.  We hope this revision addresses your concern.  If possible, we would greatly appreciate any positive feedback or a potential increase in your rating.
> >
> > Thank you once again for your thoughtful review!
> >
> >
> > Best regards,\
> > Authors of Submission 8893

---

### Official Review · Reviewer_97zR · 2024-11-04

**Soundness:** 3
**Presentation:** 2
**Contribution:** 2
**Rating:** 6
**Confidence:** 4

**Summary:**

The paper presents an improved knowledge distillation strategy for generalized category discovery (GCD), focusing on learning a better model of the known classes. Built on SimGCD, this method first introduces an auxiliary branch for the base class prediction, which distills its knowledge into the full classifier. Moreover, it incorporates a class-wise distribution regularization term to alleviate the bias toward known classes. The proposed GCD framework is evaluated on 7 datasets with comparisons to the prior arts and ablation studies on its main components.

**Strengths:**

1. The method is clearly motivated and its findings seem interesting for GCD. Based on this observation, the proposed design is reasonable.
2. The experimental results indicate that the method achieves strong performance on the benchmarks, outperforming most of the baselines.

**Weaknesses:**

1. The novelty of this work is limited. The idea of distilling known-class knowledge to improve class discovery has been explored in Gu et al, ICCV 2023, which uses a separate branch for base/known classes. Moreover, the class-wise distribution regularization technique is similar to Zhang et al. ICML 2024, which includes several properties, such as Theorem 1 in this work. While this work integrates those ingredients into SimGCD, the overall contribution seems rather limited.
2. The discussion of the related work is insufficient. Section 2 simply lists previous methods without proper comparisons to the proposed strategy.
3. There are many hyper-parameters in this method and it is unclear how they are tuned.
4. The experimental evaluation is lacking in the following aspects:
    - The baselines are insufficient. A baseline derived from NCD (Gu et al., ICCV 2023) should be included to validate the design of the AUX token.
    - The claim of this work on enhancing base classification is not well supported by the results in Table 1&2, where the base class performances are much lower than previous methods in several cases.
    - The visualization in 4.3 is not very informative as it only includes baseline methods. It should compare with the SOTA methods.
    - The ablation study shows that AUX often does not have a positive impact on the model performance. Is this component necessary for the method? It would be interesting to show the ablation results for the full model minus each component.
    - It would be more convincing to include the full results on 7 datasets for the analysis on different regularizations (ln 426) and estimated class numbers (ln 454).
5. The paper's presentation needs improvement. Some terminologies are used without proper explanation, for example:
    - ln 052: "base prediction of base unlabeled data".
    - ln 072: "pseudo-base samples"?
    - ln 199: "concentrating it with CLS ..."?
    - Theorem 2. "converges to zero": should be “equals to".

**Questions:**

Please address the comments in the weaknesses section above.

---

> ### Author Response · Authors · 2024-11-21
> **Response to Reviewer 97zR (1/6)**
>
> **Q1:** The novelty of this work is limited. The idea of distilling known-class knowledge to improve class discovery has been explored in Gu et al (ICCV 2023), which uses a separate branch for base/known classes.
>
> > Gu P, Zhang C, Xu R, et al. Class-relation knowledge distillation for novel class discovery. ICCV2023.
>
> **A1:** Thanks for bringing this paper to our attention. In fact, there are several key differences between our method and CRNCD (Gu et al., ICCV 2023), listed below.
>
> - **Different tasks.** CRNCD aims to deal with **novel class discovery (NCD)**, where all unlabeled data belong to novel classes. In contrast, our focus is on **generalized category discovery (GCD)**, where unlabeled data comprises both novel and base classes. **These two tasks are fundamentally different.** Due to the intrinsic difference between these two tasks, CRNCD demonstrates unsatisfactory performance in GCD (see **A5**).
>
> - **Different motivations for using distillation.** The distillation in CRNCD aims to improve **novel** class performance, whereas our distillation is intended to promote **base** class discrimination.
>
> - **Different training paradigms.** CRNCD adopts a **two-stage** training procedure that first trains a supervised model and then freezes it in the second stage. Contrarily, our framework is more efficient, which adopts **one-stage** training in which the main and auxiliary branches help each other simultaneously.
>
> - **Different distilled data.** While CRNCD employs distillation on **all unlabeled data**, our approach focuses on the **selective data** (i.e., pseudo-base data in our submission), which are predicted to the base classes in the main branch. The following additional experiments compare these two approaches and demonstrate that our method is more effective.
>
> | Distilled data       |   CUB200 |          |          | CIFAR100 |          |          |
> | -------------------- | --------:| -------- | -------- | --------:| -------- | -------- |
> |                      |      All | Base     | Novel    |      All | Base     | Novel    |
> | All unlabeled (CRNCD) |     66.4 | 75.4     | 61.8     |     81.8 | 83.8     | 77.8     |
> | Pseudo-base (ours)   | **69.5** | **76.4** | **65.9** | **82.8** | **84.1** | **80.1** |
>
>
> - **Different distillation weight assignments.** CRNCD adopts a **learnable weight** function to control the distillation strength. Instead, we utilize the maximum auxiliary probability as a **determined uncertainty-based weight**, which provides a **simpler and more effective** mechanism for regulating the distillation. This can be seen in the following experiments.
>
> | Weight type           |   CUB200 |          |          | CIFAR100 |          |          |
> | --------------------- | --------:| -------- | -------- | --------:| -------- | -------- |
> |                       |      All | Base     | Novel    |      All | Base     | Novel    |
> | Learnable (CRNCD)     |     65.0 | 74.5     | 60.2     |     80.9 | 82.6     | 77.5     |
> | Max. aux prob. (ours) | **69.5** | **76.4** | **65.9** | **82.8** | **84.1** | **80.1** |
>
>
> Besides the above differences, we would also like to highlight our contribution where we innovate **an AUX token to facilitate one-stage distillation**. The AUX token establishes a new feature space and effectively strengthens base discrimination with negligible extra computation.

---

> ### Author Response · Authors · 2024-11-21
> **Response to Reviewer 97zR (2/6)**
>
> **Q2:** Moreover, the class-wise distribution regularization technique is similar to Zhang et al. (ICML 2024), which includes several properties, such as Theorem 1 in this work. While this work integrates those ingredients into SimGCD, the overall contribution seems rather limited.
>
> > Yulong Zhang, Yuan Yao, Shuhao Chen, Pengrong Jin, Yu Zhang, Jian Jin, and Jiangang Lu. Rethinking guidance information to utilize unlabeled samples: A label encoding perspective. ICML2024.
>
> **A2:** Thanks for your comments. In fact, we have already made some comparisons with LERM loss (Zhang et al., ICML 2024) in our original submission (see Table 4). To better address your concern and reflect our contributions, we would like to provide some further comparisons with LERM paper as follows:
>
> - **Different loss functions.** LERM simply computes class-wise expected distribution $m_k$ for class $k$ using one view of mini-batch images and then **explicitly** enforces a one-hot distribution constraint on $m_k$ *i.e.* $m_k = e_k$ where $e_k$ is the unit vector. In contrast, our method first computes $m_k$ and $m_k'$ for view 1 and view 2, respectively. Then the proposed CDR drives $m_k$ and $m_k'$ to become similar, **implicitly** guiding $m_k$ toward a one-hot distribution. Moreover, LERM overlooks the inter-class relationship, and the explicit regularization often leads to overconfidence. To solve this issue, our CDR l**everages probability consistency to implicitly increase prediction confidence.**  Therefore, LERM and our method employ fundamentally different regularization mechanisms.
>
> - **Significant improvement.** Due to the above advantages of our method, Table 4 confirms that our CDR outperforms LERM significantly. Specifically, our CDR loss achieves an accuracy improvement of 2.3% on CIFAR100, from 80.5% (LERM) to 82.8% (CDR).
>
> - **Theorem 1 is not our contribution, but Theorem 2 is.** While Theorem 1 holds both for LERM and our CDR, we do not regard Theorem 1 as our contribution (as we have provided its reference in the paragraph above Theorem 1 in our original submission). In fact, it serves as a **precondition** for our Theorem 2, which represents the **fundamental insight** of our method. We have added a clear citation beside the Theorem 1 in our revision.
>
>
>
> In addition, we would like to emphasize again that our key novelty stems from the efficient **one-stage distillation framework** and the **refined class-specific distribution regularization**.
>
>
>
> ---
>
> **Q3:** The discussion of the related work is insufficient. Section 2 simply lists previous methods without proper comparisons to the proposed strategy.
>
> **A3:** Thanks for your constructive suggestions. In our latest revised submission, we have added the two related papers mentioned in **A1** and **A2**, along with the corresponding discussions in Section 2 as follows:
>
>
> - Prevailing semi-supervised learning methods assume that unlabeled data belong to known classes and widely adopt threshold-based pseudo-label learning during training. However, this mechanism **faces significant limitations** when unlabeled data include unknown class samples.
> -  Current SOTA NCD methods are primarily built on the UNO [1] framework and they assume all unlabeled data belong to novel classes. Besides, these methods often utilize the Optimal-Transport (OT) algorithm to dynamically cluster unlabeled data. However, OT achieves unsatisfactory novel class performance when unlabeled data involves known class samples. Consequently, **current NCD methods are unsuitable for the GCD task**. Moreover, CRNCD [2] introduces a two-stage class-relationship distillation approach to improve novel-class performance. However, we observe that this distillation shows inferior performance on GCD. Unlike CRNCD [2], we propose a novel one-stage distillation method tailored for GCD.
> -  While parametric-based methods achieve great GCD performance, they often **suffer degraded base discrimination**. To address this issue, we propose a Reciprocal Learning Framework (RLF) that introduces an auxiliary branch specialized in base classification. RLF provides more reliable base pseudo-labels and effectively strengthens base performance with negligible extra computation cost. Combined with class-wise distribution regularization, our method achieves superior performance than SOTA GCD methods.
>
> *[1] Fini E, Sangineto E, Lathuilière S, et al. A unified objective for novel class discovery. ICCV2021. \
> [2] Gu P, Zhang C, Xu R, et al. Class-relation knowledge distillation for novel class discovery. ICCV2023.*

---

> ### Author Response · Authors · 2024-11-21
> **Response to Reviewer 97zR (3/6)**
>
> **Q4:** There are many hyper-parameters in this method and it is unclear how they are tuned.
>
> **A4:** Thanks for pointing this out. First of all, we would like to clarify that most of the hyper-parameters are simply **adopted from previous works**, such as $\lambda = 0.35$, $\epsilon = 2.0$, while our method only introduces two additional hyper-parameters: $\alpha$ and $\beta$.
>
> **For those hyper-parameters adopted from previous works**, we follow the same configuration in their original papers (i.e., SimGCD [1] and LegoGCD [2]) as reasonable default choices. We also validate these hyper-parameters and they indeed achieve the optimal result, which is consistent with existing works.
>
> -  Specifically, the balancing weight $\lambda \in [0, 1]$, and the default $\lambda = 0.35$. The ablation study results on CUB200 are as follows:
>
> | $\lambda$ | 0.0  | 0.1  | 0.3  | 0.35 | 0.4  | 0.6  | 1.0  |
> | ---- | ---- | ---- | ---- | -------- | ---- | ---- | ---- |
> | Acc. | 65.3 | 67.9 | 68.6 | **69.5** | 66.5 | 62.1 | 39.7 |
>
> - Hyper-parameter $\epsilon$ is used for weighting mean-entropy regularization ($H$), and the default value is set to $2$ on CUB200. The ablation results are as follows:
>
> | $\epsilon$ | 0.0  | 1.0  | 2.0  | 3.0  | 4.0  |
> | ---- | ---- | ---- | -------- | ---- | ---- |
> | Acc. | 41.7 | 53.3 | **69.5** | 64.7 | 56.7 |
>
> **For the two new hyper-parameters $\alpha$ and $\beta$**, where $\alpha$ adjusts distillation strength and $\beta$ controls regularization strength, we conduct experiments in our submission to analyze the effect of these two hyper-parameters $\alpha$ and $\beta$.
>
> - The results have been shown in Figure 5 of our submission, indicating that they are **not sensitive** to the final performance. Based on the above observations, we choose to use the value of $0.5$ for both $\alpha$ and $\beta$.
>
>
> *[1] Wen X, Zhao B, Qi X. Parametric classification for generalized category discovery: A baseline study. ICCV2023.\
> [2] Cao X, Zheng X, Wang G, et al. Solving the catastrophic forgetting problem in generalized category discovery. CVPR2024.*
>
> ---
>
> **Q5：** The baselines are insufficient. A baseline derived from NCD (Gu et al., ICCV 2023) should be included to validate the design of the AUX token.
>
> > Gu P, Zhang C, Xu R, et al. Class-relation knowledge distillation for novel class discovery. ICCV2023.
>
> **A5：** Thanks for your constructive suggestion. Since CRNCD (Gu et al., ICCV 2023) is tailored for the NCD task, it cannot directly apply to the GCD task considered in our paper. To this end, we have **incorporated the core idea of CRNCD** into the GCD task and included the reproduced results in Tables 1 and 2 in our revised submission. Below we provide a sketch result (please see full results in our revision):
>
> |          | CIFAR10  | CIFAR100 | ImageNet-100 | CUB      | Stanford cars | Aircraft | Herbarium-19 | Avg.     |
> | :--------: | :--------: | :--------: | :------------: | :--------: | :-------------: | :--------: | :------------: | :--------: |
> | CRNCD (ICCV 2023) | 96.9     | 80.3     | 81.4         | 62.7     | 54.1          | 54.4     | 41.3         | 67.3     |
> | Ours     | **97.4** | **82.8** | **86.5**     | **69.5** | **61.1**      | **60.6** | **46.4**     | **72.0** |
>
>
> The table above shows the overall accuracy comparison across seven GCD datasets between CRNCD and our method. The result demonstrates that **CRNCD is unsuitable in GCD**, consistently underperforming compared to our method across all datasets. Notably, our method outperforms CRNCD by 7.0% on Stanford cars dataset and achieves an average improvement of 4.7% on 7 datasets.

---

> ### Author Response · Authors · 2024-11-21
> **Response to Reviewer 97zR (4/6)**
>
> **Q6:** The claim of this work on enhancing base classification is not well supported by the results in Table 1&2, where the base class performances are much lower than previous methods in several cases.
>
> **A6:** Thanks for your valuable feedback.
>
> First, we would like to clarify that our method achieves **superior base accuracy** compared to other parametric-based methods, such as SimGCD [1] and LegoGCD [2]. In these cases, the results **align with our motivation**. Besides, our approach also achieves leading oracle base class performance shown in Figure 1 and Appendix A.5.
>
> Second, while our base class accuracy may lag behind a few methods belonging to other taxonomies, we attribute this to the **essential limitation of parametric-based methods**, whose losses include **entropy mean regularization**. Concretely, this regularization encourages diverse prediction, and thus partial base class data would be misclassified as the novel [2]. Methods like GPC [3] and UNO [4], which are not parametric-based methods, avoid such regularization, and naturally exhibit higher base accuracy but often sacrifice novel performance.
>
> Furthermore, the reported base accuracy is adjusted by the Hungarian algorithm [5], and is **not directly equivalent to the true base accuracy.** For instance, while SPTNet [6] shows a higher reported base accuracy (79.2% vs. 76.6%), the true base accuracy is similar (81.4% vs. 82.1%), based on results reproduced from  SPTNet's official checkpoint [7].
>
> The above discussion has been added to our revision to better support the motivation of this work.
>
> *[1] Wen X, Zhao B, Qi X. Parametric classification for generalized category discovery: A baseline study. ICCV2023.\
> [2] Cao X, Zheng X, Wang G, et al. Solving the catastrophic forgetting problem in generalized category discovery. CVPR2024.\
> [3] Zhao B, Wen X, Han K. Learning semi-supervised gaussian mixture models for generalized category discovery. ICCV2023.\
> [4] Fini E, Sangineto E, Lathuilière S, et al. A unified objective for novel class discovery. ICCV2021.\
> [5] Kuhn H W. The Hungarian method for the assignment problem. Naval research logistics quarterly, 1955.\
> [6] Wang H, Vaze S, Han K. SPTNet: An efficient alternative framework for generalized category discovery with spatial prompt tuning.\
> [7] https://github.com/Visual-AI/SPTNet*
>
> ---
>
> **Q7:** The visualization in 4.3 is not very informative as it only includes baseline methods. It should compare with the SOTA methods.
>
> **A7:** Thanks for your thoughtful comment. To present a more informative visualization, we add the most recent work LegoGCD [1] (CVPR 2024) as a comparison and our approach still performs the best clustering visualization result. **Please refer to Figure 4 in the revised submission.**
>
> In brief, the t-SNE for pre-trained DINO [2] shows **random clustering** results, primarily due to the significant domain gap between the ImageNet and Aircraft datasets. Meanwhile, both  LegoGCD [1] and SimGCD [3] exhibit unsatisfactory feature clustering. Specifically, some intra-class features of SimGCD are **spread out** in the feature space, while LegoGCD forms **two separate clusters for a single class**. In contrast, our model reveals **distinct clusters** corresponding to different categories. The visualization comparison validates the superior feature representation of our model.
>
> *[1] Cao X, Zheng X, Wang G, et al. Solving the catastrophic forgetting problem in generalized category discovery. CVPR2024.\
> [2] Caron M, Touvron H, Misra I, et al. Emerging properties in self-supervised vision transformers. ICCV2021.\
> [3] Wen X, Zhao B, Qi X. Parametric classification for generalized category discovery: A baseline study. ICCV2023.*

---

> ### Author Response · Authors · 2024-11-21
> **Response to Reviewer 97zR (5/6)**
>
> **Q8:** Is AUX component necessary for the method? It would be interesting to show the ablation results for the full model minus each component.
>
> **A8:** Thanks for your valuable comments. In fact, AUX is indispensable in our method. Below, we give a detailed statement about AUX and provide additional ablation results:
>
> - **AUX Creates a complementary feature space.** The AUX component is crucial to our framework as it establishes an independent feature space that **complements the main CLS branch**. This design enables AUX to enhance base performance through the distillation loss, completing the one-stage reciprocal learning.
>
> - **Removing AUX leads to a significant decrease in performance.** We also quantitatively **validate the necessity** of AUX by temporarily removing this component. When AUX is removed, the base and full classifiers are forced to share the same feature space. This shared space results in a significant performance drop, as shown in the table below. The metric OB in the table denotes oracle base class accuracy to evaluate pseudo-label quality.
>
> - **Additional ablations of each component.** To provide a comprehensive analysis, we have extended the ablation experiment in Table 3 of the submission to include the impact of removing each component from the full model. These results clearly highlight the importance of each component, including AUX, Distill, and CDR. **Both AUX and Distill contribute to improved base performance and CDR effectively enhances novel performance.**
>
>
> |                  |    CIFAR100 |          |          |          |      CUB200 |          |          |          |
> | ---------------- | -----------:| -------- | -------- | -------- | -----------:| -------- | -------- | -------- |
> |                  |         All | Base     | Novel    | OB       |         All | Base     | Novel    | OB       |
> | Ours w/o AUX     |  81.0 (↓1.8) | 81.9     | 79.2     | 84.9     | 65.9 (↓3.6) | 73.8     | 61.9     | 85.6     |
> | Ours w/o Distill | 81.6 (↓1.2) | 82.7     | 79.2     | 85.5     | 67.6 (↓1.9) | 71.7     | 65.6     | 85.8     |
> | Ours w/o CDR     | 80.8 (↓2.0) | 83.8     | 74.7     | **86.2** |  66.5 (↓3.0) | 75.9     | 61.8     | 87.2     |
> | Ours             |    **82.8** | **84.1** | **80.1** | 86.0     |    **69.5** | **76.4** | **65.9** | **87.3** |
>
> ---
>
> **Q9:** It would be more convincing to include the full results on 7 datasets for the analysis of different regularizations (ln 426) and estimated class numbers (ln 454).
>
> **A9:** Thanks for your thoughtful suggestion. We have extended the analysis experiment of different regularizations and estimated class numbers on other datasets.
>
> For comparing to **different regularizations**, we are here to report the main result of "All accuracy", and the detailed performance can be found in the updated submission. Note that, PDR is the modification of CDR, denoting pair-wise class-distribution regularization.
>
>
> | Regularizations | CIFAR10  | ImageNet-100 |  Scars   | AirCraft | Herbarium-19 |
> | --------------- |:--------:|:------------:|:--------:|:--------:|:------------:|
> | ENT [1]         |   96.5   |     81.5     |   52.1   |   51.7   |     40.9     |
> | MCC [2]         |   97.2   |     83.6     |   56.3   |   56.3   |     44.5     |
> | BNM [3]         |   97.0   |     83.2     |   55.4   |   55.8   |     44.2     |
> | LERM [4]        |   97.1   |     85.1     |   56.6   |   56.2   |     44.0     |
> | PDR (modification of CDR)             |   97.2   |     86.1     |   58.7   |   57.2   |     44.8     |
> | Ours       | **97.4** |   **86.5**   | **61.1** | **60.6** |   **46.5**   |
>
>
> In addition, we also provide extra experiments on the **estimated class numbers**. Since GCP accurately estimates the category number for CIFAR100 and CIFAR10, we show the comparison result under CMS [5] estimation.
>
>
> |         | CIFAR100 |  Scars   | Aircraft | Herbarium-19 |
> | ------- |:--------:|:--------:|:--------:|:------------:|
> | CMS [5] |   79.6   |   51.7   |   55.2   |     37.4     |
> | Ours    | **80.0** | **56.6** | **58.2** |   **46.0**   |
>
>
> From the above results, our CDR **consistently outperforms** other regularization methods on various datasets. Additionally, our approach **performs better** than CMS when dealing with estimated class numbers.
>
> *[1] Yves Grandvalet and Yoshua Bengio. Semi-supervised learning by entropy minimization. NeurIPS2004.\
> [2] Jin Y, Wang X, Long M, et al. Minimum class confusion for versatile domain adaptation. ECCV2020.\
> [3] Cui S, Wang S, Zhuo J, et al. Towards discriminability and diversity: Batch nuclear-norm maximization under label insufficient situations. CVPR2020.\
> [4] YZhang Y, Yao Y, Chen S, et al. Rethinking guidance information to utilize unlabeled samples: A label encoding perspective. ICML2024.\
> [5] Choi S, Kang D, Cho M. Contrastive mean-Shift learning for generalized category discovery. CVPR2024.*

---

> ### Author Response · Authors · 2024-11-21
> **Response to Reviewer 97zR (6/6)**
>
> **Q10:** The paper's presentation needs improvement. Some terminologies are used without proper explanation, for example: ln 052: "base prediction of base unlabeled data". ln 072: "pseudo-base samples" ln 199: "concentrating it with CLS ..."? Theorem 2. "converges to zero": should be “equals to".
>
> **A10:** Thank you for your insightful feedback. We truly appreciate your attention to detail. Based on your comments, we have carefully revised the relevant sections in the updated version of our manuscript to improve clarity and accuracy.

---

### Author Response · Authors · 2024-11-24
**A Summary of Paper Updates**

We thank all reviewers for the constructive suggestions, which help make this work more complete. Following their suggestions, we have fixed the typos and made the following major updates to the paper:

- **Section 1**: Add a detailed explanation of the term “pseudo-base.”
- **Section 2**: Expand the references and discussions to enrich the related work.
- **Section 3**:
    - Clarify why the InfoNCE loss is not used.
    - Conduct a deeper analysis of the inferior discrimination observed in parametric-based methods.
    - Provide an explanation for the variable $m_k$.
- **Section 4**:
    - Include additional comparative methods in the main results.
    - Include LegoGCD [1] for more informative visualizations.
    - Extend the ablation study to analyze the impact of removing each component of the full model and validate CDR solely on the main branch.
    - Expand experiments to investigate the effects of various regularizations and estimated category numbers across more datasets.
- **Appendix A.7**: Add discussions related to CRNCD [2].
- **Appendix A.8**: Include additional results for different regularizations.
- **Appendix A.9**:  Include additional results for estimated category numbers.
- **Appendix A.10**: Introduce comparisons leveraging the DINOv2 backbone.
- **Appendix A.11**: Introduce further analysis of $H$.

*[1] Cao X, Zheng X, Wang G, et al. Solving the catastrophic forgetting problem in generalized category discovery. CVPR2024.*\
*[2] Gu P, Zhang C, Xu R, et al. Class-relation knowledge distillation for novel class discovery. ICCV2023.*

---

### Meta-Review · Area_Chair_YbsF · 2024-12-17

**Metareview:**

This paper considers the problem of Generalized Category Discovery (GCD)  and focuses on the drawback of recent parametric-based methods. To solve this, this paper proposes a Reciprocal Learning Framework (RLF), which can mitigate inferior base discrimination caused by unreliable self-supervision.

The paper was reviewed by four reviewers. The main the strengths recognized by the reviewers are: 1) interesting findings; 2) strong performance; and 3) well-written.

Despite these, the reviewers also raised concerns and drawbacks of this paper. The authors have provided a rebuttal to solve them accordingly. All the reviewers have attended the discussion-phase, in which three of them are satisfied with the responses and finally gave positive scores to this paper. However, the Reviewer hpPJ still has the concerns about the 1) inconsistent motivation throughout the paper, in which the motivation should be consistently referenced throughout the paper, including in the experimental section; and 2) the root cause of bias has not been fully solved, in which ( H ) is the primary source of bias and addressing the bias from the the perspective of (H) would be more appropriate and reasonable. Given these two concerns, the Reviewer hpPJ kept a negative score to this paper. The AC agrees with the Reviewer hpPJ that this paper should provide a more consistent clarification of the motivation throughout the paper and clearly analyze the underlying reason of the bias. On the other hand, no one champions the acceptance for this paper. Given the above considerations, the AC thinks the current version of this paper is not ready to be published in ICLR and regrets to recommend rejection to this paper.

**Additional Comments On Reviewer Discussion:**

The authors have provided a rebuttal. One reviewer raised the score from 5 to 6 while the others kept their original ratings. The paper finally got one borderline rejection and three borderline acceptance. The main remaining concerns are 1) inconsistent motivation throughout the paper and 2) the root cause of bias has not been fully solved. The AC agrees that these two concerns are important for this paper and should be clearly solved before publication.

---

### Decision · Program_Chairs · 2025-01-22

Reject